



# Modeling Diurnal Variation of SOA Formation via Multiphase Reactions of Biogenic Hydrocarbons

Sanghee Han[1], Myoseon Jang[1]

[1]Department of Environmental Engineering Science, University of Florida, Gainesville, Florida, USA

5  *Correspondence to:* Myoseon Jang (mjang@ufl.edu)

**Abstract.**

The daytime oxidation of biogenic hydrocarbons is attributed to both OH radicals and $O_3$, while nighttime chemistry is dominated by the reaction with $O_3$ and $NO_3$ radicals. Here, the diurnal pattern of Secondary Organic Aerosol (SOA) originating from biogenic hydrocarbons was intensively evaluated under varying environmental conditions (temperature, humidity, 10  sunlight intensity, $NO_x$ levels, and seed conditions) by using the UNIfied Partitioning Aerosol phase Reaction (UNIPAR) model, which comprises multiphase gas-particle partitioning and in-particle chemistry. The oxidized products of three different hydrocarbons (isoprene, α-pinene, and β-caryophyllene) were predicted by using near explicit gas mechanisms for four different oxidation paths (OH, $O_3$, $NO_3$, and $O(^3P)$) during day and night. The gas mechanisms implemented the Master Chemical Mechanism (MCM v3.3.1), the reactions that formed low volatility products via peroxy radical ($RO_2$) autoxidation, 15  and self- and cross-reactions of nitrate-origin $RO_2$. In the model, oxygenated products were then classified into volatility-reactivity base lumping species, which were dynamically constructed under varying $NO_x$ levels and aging scales. To increase feasibility, the UNIPAR model that equipped mathematical equations for stoichiometric coefficients and physicochemical parameters of lumping species was integrated with the SAPRC gas mechanism. The predictability of the UNIPAR model was demonstrated by simulating chamber-generated SOA data under varying environments day and night. Overall, the SOA 20  simulation decoupled to each oxidation path indicated that the nighttime isoprene SOA formation was dominated by the $NO_3$-driven oxidation, regardless of $NO_x$ levels. However, the oxidation path to produce the nighttime α-pinene SOA gradually transited from the $NO_3$-initiated reaction to ozonolysis as $NO_x$ levels decreased. For daytime SOA formation, both isoprene and α-pinene were dominated by the OH-radical initiated oxidation. The contribution of the $O(^3P)$ path to all biogenic SOA formation was negligible in daytime. Sunlight during daytime promotes the decomposition of oxidized products via photolysis 25  and thus, reduces SOA yields. Nighttime α-pinene SOA yields were significantly higher than daytime SOA yields, although the nighttime α-pinene SOA yields gradually decreased with decreasing $NO_x$ levels. For isoprene, nighttime chemistry yielded higher SOA mass than daytime at the higher $NO_x$ level (isoprene/$NO_x$ > 5 ppbC/ppb). The daytime isoprene oxidation at the low $NO_x$ level formed epoxy-diols that significantly contributed SOA formation via heterogeneous chemistry. For isoprene and α-pinene, daytime SOA yields gradually increased with decreasing $NO_x$ levels. The daytime SOA produced more highly 30  oxidized multifunctional products and thus, it was generally more sensitive to the aqueous reactions than the nighttime SOA. β-Caryophyllene, which rapidly oxidized and produced SOA with high yields, showed a relatively small variation in SOA yields from changes in environmental conditions (i.e., $NO_x$ levels, seed conditions, and diurnal pattern), and its SOA formation was mainly attributed to ozonolysis day and night. To mimic the nighttime α-pinene SOA formation under the polluted urban atmosphere, α-pinene SOA formation was simulated in the presence of gasoline fuel. The simulation suggested the growth of 35  α-pinene SOA in the presence of gasoline fuel gas by the enhancement of the ozonolysis path under the excess amount of ozone, which is typical in urban air. We concluded that the oxidation of the biogenic hydrocarbon with $O_3$ or $NO_3$ radicals is a source to produce a sizable amount of nocturnal SOA, despite of the low emission at night.



## 1 Introduction

Organic aerosol in the ambient air has been a well-known factor to impact human health (Pye et al., 2022;Mauderly and Chow, 2008) and climate change (Tsigaridis and Kanakidou, 2018;Kanakidou et al., 2005). A large portion of organic aerosol is secondary organic aerosol (SOA) produced from the oxidation process of hydrocarbons (HCs), emitted from both biogenic and anthropogenic sources (Hallquist et al., 2009;Jimenez et al., 2009). In particular, the global emission of biogenic HCs, which is more than two-thirds of the total HC emissions, dominate over that of anthropogenic HCs by an order magnitude

(Guenther et al., 1995;Goldstein and Galbally, 2007;Sindelarova et al., 2014). These biogenic HCs contain olefinic (C=C) bonds that are highly reactive towards various oxidants (i.e., OH radicals, $NO_3$ radicals, and $O_3$) (Atkinson and Arey, 2003). Furthermore, the SOA from the oxidation of biogenic HCs is considerable in a global budget of SOA. For example, the annual global SOA production rates from monoterpene and isoprene is more than 50% of the global SOA formation, 19.9 and 19.6 Tg (SOA) $a^{-1}$, respectively (Kelly et al., 2018).

In the daytime, a large amount of biogenic HC is oxidized mainly with OH radicals and $O_3$ to form a considerable SOA burden (Zhang et al., 2018;Carlton et al., 2009;Sakulyanontvittaya et al., 2008;Barreira et al., 2021). The photooxidation of $NO_x$ enhances the production of $O_3$ and regenerate OH radicals, increasing the consumption of biogenic HCs and SOA formation. In nighttime, however, the oxidation of biogenic HCs with the OH radical is minimized, while that of biogenic HCs is processed dominantly by $O_3$ and $NO_3$ radicals. The $O_3$ that is generated in daytime is persistent at nighttime. A $NO_3$ radical that forms

via the reaction of $O_3$ with $NO_2$ can also be sustainable in nighttime. Thus, the oxidation pathways of biogenic HCs can change diurnally with different $NO_x$ levels, humidity, and temperature, and ultimately influence SOA formation. For example, the oxidation of isoprene with the $NO_3$ radical can rapidly produce nitrate containing products, such as $C_5$-nitroxycarbonyl and $C_5$-hydroxynitrate, up to 80% of gas products from the isoprene-$NO_3$ oxidation (Kwok et al., 1996;Barnes et al., 1990). Numerous studies recently have also shown the important role of the $NO_3$ radical on the production of SOA, suggesting the

emission of $NO_x$ from human activities increases the biogenic SOA mass (Ng et al., 2008;Bonn and Moorgat, 2002;Jaoui et al., 2013;Rollins et al., 2012). Furthermore, the variation of the inorganic seed effects on SOA formation can be significant by different oxidation pathways, which produce different product distributions.

In current air quality models, the partitioning-based SOA models semi-empirically established a relationship between the absorbing organic matter (OM) concentration and the SOA yields by using a simple model parameters for two (Odum et al.,

1996) or more surrogate products (Donahue et al., 2006). The oxidation of biogenic HCs was approached by the reaction with four major oxidants: OH radicals, $NO_3$ radicals, $O_3$, and $O(^3P)$. Consequently, the biogenic SOA formation is predicted with the surrogate products originating from these four major oxidations. However, the gas phase reactions were not additive due to the various cross reactions. For example, the 1st generation of oxidation products initiated by the ozone mechanism can react with the OH radical. The product distribution originating from a specific oxidant can also influenced by the $NO_x$ level and

atmospheric gas aging.

Numerous studies have shown the importance of the aerosol phase reaction, yielding the non-volatile species or oligomeric matter, of reactive organic species (i.e., aldehydes and epoxides) in aerosol phase (Jang et al., 2002;Woo et al., 2013;Altieri et al., 2006;Ervens et al., 2004;Liggio et al., 2005). The typical partitioning-based models include organic-phase oligomerization of organic species, but they do not fully treat the SOA formation via the aqueous reactions in the presence of inorganic salted

aqueous phase. To predict SOA formation via the multiphase reaction of HCs, the UNIfied Partitioning Aerosol-phase Reaction (UNIPAR) model has been developed (Beardsley and Jang, 2016;Im et al., 2014;Zhou et al., 2019). UNIPAR streamlines the gas oxidation integrated with explicit mechanisms, multiphase partitioning, and aerosol-phase reactions in both organic and inorganic phases. This model has been demonstrated by the SOA produced from various aromatic HCs (Im et al., 2014;Zhou et al., 2019;Han and Jang, 2022), monoterpenes (Yu et al., 2021), and isoprene (Beardsley and Jang, 2016).





In this study, to predict the diurnal pattern of biogenic SOA formation, the UNIPAR model was extended to biogenic SOA formation at nighttime. Lumping species and their stoichiometric coefficient and physicochemical parameters from the explicit gas mechanisms were individually generated from the four different oxidation pathways with OH radicals, $O_3$, $NO_3$ radicals, and $O(^3P)$. To improve the feasibility, the UNIPAR model was integrated with the SAPRC07TC gas mechanism (UNIPAR-SAPRC), providing the HC consumption by each oxidant. The resulting model allows the prediction of biogenic SOA in day

and night with the dynamically controlled product distribution by oxidation of gas simulation. The potential SOA yield of biogenic HCs via four different oxidation paths were investigated by simulation of the UNIPAR-SAPRC model and applied to clarify the diurnal patterns in biogenic SOA formation under varying $NO_x$ levels, temperature, and seed conditions. Additionally, the UNIPAR-SAPRC model was utilized to analyze the impact of anthropogenic emissions on biogenic SOA formations.

**2 Chamber experiment**

The chamber experiments to produce SOA from the oxidation process of biogenic HCs were conducted in the University of Florida Atmospheric PHotochemical Outdoor Reactor (UF-APHOR) chamber located on the rooftop of Black Hall (29.64 °, -82.34 °) at the University of Florida, Gainesville, Florida. The detailed configuration of the UF-APHOR and the experimental procedures were previously reported (Beardsley and Jang, 2016;Im et al., 2014;Zhou et al., 2019). In brief, UF-APHOR

chamber is a dual chamber (52 $m^3$ (East) + 52 $m^3$ (West)) made with Fluorinated Ethylene Propylene (FEP) Teflon film. For daytime experiments, the injection was done before sunrise, and the experiments started at sunrise, conducted for 10 to 12 hours. The $NO_x$ levels were controlled by NO injected from the 2% NO cylinder under the air flow and classified into high $NO_x$ (HC/$NO_x$ < 5.5 ppbC/ppb) and low $NO_x$ level (HC/$NO_x$ > 5.5 ppbC/ppb) based on the initial concentration of HC and NO. Inorganic seed aerosols (sulfuric acid, (SA), wet ammonium sulfate, (wet-AS), and dry ammonium sulfate (dry-AS)) were

injected into the chamber to evaluate the effects of wet inorganic seed on SOA formation. For nighttime experiments, the injection and experiments began after sunset to avoid photochemical reaction, and experiments were conducted for 3 to 5 hours. $O_3$ was injected first into the chamber by using the $O_3$ generator, and $NO_x$ condition was controlled by $NO_2$. $NO_2$ was injected from the 2% $NO_2$ cylinder into the chamber under the air flow. Nighttime biogenic SOA formation was observed under the three different $NO_x$ levels (i.e., $O_3$ only, low $NO_x$ (HC/$NO_x$ > 5.5 ppbC/ppb) and high $NO_x$ level (HC/$NO_x$ < 5.5

ppbC/ppb)). Three different biogenic HCs (isoprene ($C_5H_8$, 99%, Aldrich), α-pinene ($C_{10}H_{16}$, 98% Aldrich), and β-caryophyllene ($C_{15}H_{24}$, >90%, TCI)) were injected as a last step of the injection before experiments started.

The concentration of HCs and $CCl_4$ were monitored using a Gas Chromatography–Flame Ionizer Detector (Agilent, model 7820A) (GC-FID). The HC concentration detected by GC–FID determined HC consumption in the chamber during the experiment. The concentration of $CCl_4$ measured by GC-FID was monitored as a function of time to obtain the dilution factor

of the chamber during the experiment. The concentration of $O_3$ was monitored with a photometric ozone analyzer (Teledyne, model 400E and 2B Technologies, model 106-L, M). $NO_x$ concentration was monitored by using a chemiluminescence NO/$NO_2$ analyzer (Teledyne, model 200E) and photometric $NO_x$ analyzer (2B Technologies, model 405 nm). The inorganic ion ($SO_4^{2-}$ and $NH_4^+$) and organic carbon (OC) concentrations of aerosol were in situ monitoring by the Particle-Into-Liquid-Sampler (Applikon, ADI 2081), coupled with Ion Chromatography (Metrohm, 761Compact IC) (PILS–IC), and an OC/EC

carbon aerosol analyzer (Sunset Laboratory, Model 4), respectively. The Scanning Mobility Particle Sizer (SMPS, TSI, Model 3080) integrated with a condensation nuclei counter (TSI, Model 3025A and Model 3022) was used to measure the particle volume concentration over the course of experiment. An Aerosol Chemical Speciation Monitor (ACSM, Aerodyne Research Inc.) observed the composition ($SO_4^{2-}$, $NO_3^-$, $NH_4^+$, and organic) of aerosol to compare with the data obtained from OC and PILS–IC for the accurate measurement. The relative humidity and temperature were monitored in the UF-APHOR and applied

to the simulation, and the sunlight intensity was measured by Total Ultraviolet Radiometer (EPLAB, TUVR). Aerosol acidity


(mol/L of aerosol) was examined by Colorimetry integrated in the Reflectance UV–visible spectrometer (CRUV) (Li and Jang, 2012;Jang et al., 2020). The details of the experimental conditions are summarized in Table 1.

## 3 Model descriptions

In the model, the oxidation products from each biogenic HC were predicted by using explicit mechanisms for each oxidant
(OH radicals, $O_3$, $NO_3$ radicals, and $O(^3P)$). The simulated gas products were classified into the 51 lumping species ($i$) according to their volatility and reactivity in aerosol phase. The stoichiometric coefficients ($\alpha_i$) array and physicochemical parameters ($p^{\circ}_{L,i}$, $MW_i$, $O:C_i$, and $HB_i$) of the lumping species ($i$) were determined by the near-explicit gas mechanisms. In order to increase the suitability, the UNIPAR model was coupled with the conventional ozone mechanisms, such as SAPRC07TC as seen in Fig. 1 (UNIPAR-SAPRC). The UNIPAR-SAPRC model was simulated under the Dynamically Simple
Model of Atmospheric Chemical Complexity (DSMACC) (Emmerson and Evans, 2009) integrated with the Kinetic PreProcessor (KPP) (Damian et al., 2002). The HC consumption obtained from SAPRC, $\alpha_i$, and physicochemical parameters of lumping species ($i$) were then applied to produce SOA mass ($OM_T$) via gas–particle partitioning ($OM_P$) and heterogeneous reactions ($OM_{AR}$) in both organic and inorganic phases. For α-pinene and β-caryophyllene, the SOA formation in the presence of salted aqueous solution was simulated under the assumption of the liquid-liquid phase separation (LLPS) between the
organic and inorganic phase. The simulation of this study is limited to LLPS and thus, the isoprene SOA formation in the presence of inorganic aerosol was excluded. The water content in isoprene SOA, which is very hydrophilic, was estimated with 1/3 of hygroscopicity of ammonium sulfate (Beardsley and Jang, 2016). The details of the UNIPAR model were described in the following sections.

### 3.1 Generation of lumping species from the Explicit Gas Mechanisms

The gas-phase oxidation of three biogenic HCs (isoprene, α-pinene, and β-caryophyllene) of this study was explicitly processed by using the Master Chemical Mechanism (MCM v3.3.1) (Saunders et al., 2003;Jenkin et al., 2012;Jenkin et al., 2015) to generate lumping species and their model parameters. Additionally, the recently identified oxidation mechanisms that can yield low volatile products were also integrated with MCM. For example, the Peroxy Radical Autoxidation Mechanism (PRAM) (Roldin et al., 2019) that forms the highly oxygenated organic molecule (HOM) (Molteni et al., 2019) and the
accretion reaction to form ROOR from the $RO_2$ (Bates et al., 2022;Zhao et al., 2021) were added. Furthermore, the oxidation process of biogenic HCs by $O(^3P)$ (Paulson et al., 1992;Alvarado et al., 1998) was included to fulfill the oxidation mechanism in the current regional model. The additional mechanisms are shown in the Sect. S1.2. The oxidation-path dependent lumping parameters were generated by individually processing the reaction of biogenic HCs with four major oxidants (OH radicals, $O_3$, $NO_3$ radicals, and $O(^3P)$). After a biogenic HC is oxidized with individual oxidants, the further oxidation of the 1st generation
product was allowed to react with any oxidant. For instance, the 1st generation ozonolysis products of biogenic HC can react with OH radicals or $NO_3$ radicals.

The resulting oxygenated products from explicit gas mechanism were classified into eight levels of the vapor pressure ($P^{\circ}_{L,i}$) (1–8: $10^{-8}$, $10^{-6}$, $10^{-5}$, $10^{-4}$, $10^{-3}$, $10^{-2}$, $10^{-1}$, and 1 mmHg) and six levels of the aerosol–phase reactivity scale ($R_i$): very fast (VF), fast (F), medium (M), slow (S), partitioning only (P), and multi-alcohol (MA), as well as three additional reactive species
(glyoxal, methylglyoxal, and epoxydiols). The physicochemical parameters ($p^{\circ}_{L,i}$, $MW_i$, $O:C_i$, and $HB_i$) of lumping species ($i$) are determined based on the group contribution (Stein and Brown, 1994) and unified into one array for each HC. $\alpha_i$ for each oxidation pathway was estimated by the predetermined mathematical equation originated from the explicit mechanism as a function of HC/$NO_x$ (ppbC/ppb) level and aging factor ($f_A$). $f_A$ is calculated from the concentration of HC, $RO_2$, and $HO_2$ (Zhou et al., 2019). Additionally, the product distribution leading to $\alpha_i$ can be impacted by the ozone to $NO_x$ ratio, particularly





at nighttime. In our study, the predetermined mathematical equation to construct $\alpha_i$ at night was determined at the constant

HC (ppb) to ozone (ppb) ratio as 0.25.

**3.2 SOA growth via gas-particle partitioning**

In this model, the gas–particle partitioning of oxidation products are assumed as an equilibrium partitioning process based on

the absorptive partitioning theory (Pankow, 1994). It assumes that the gas–particle partitioning instantaneously reaches

equilibrium to distribute the gas products into the gas ($C_{g,i}$), organic ($C_{or,i}$) and inorganic phases ($C_{in,i}$). The partitioning

coefficient of $i$ into the organic phase ($K_{or,i}$, m³ µg⁻¹) is determined by the traditional absorptive partitioning theory (Pankow,

1994) as follows:

$$K_{or,i} = \frac{7.501RT}{10^9 MW_{or}\gamma_{or,i}p_{L,i}^\circ}, \tag{1}$$

where $MW_{or}$ (g mol⁻¹) is the molecular weight of OM$_T$, $R$ (8.314 J mol⁻¹ K⁻¹) is the ideal gas constant, and $T$ (K) is the

temperature. $\gamma_{or,i}$ is the activity coefficient of $i$ in organic phase and assumed as unity. The partitioning coefficient of $i$ into

the inorganic phase ($K_{in,i}$, m³ µg⁻¹) is also calculated as the traditional absorptive partitioning theory:

$$K_{in,i} = \frac{7.501RT}{10^9 MW_{in}\gamma_{in,i}p_{L,i}^\circ}, \tag{2}$$

where $MW_{in}$ (g mol⁻¹) is the averaged molecular weight of inorganic aerosol, and $\gamma_{in,i}$ is the activity coefficient of $i$ in

inorganic phase. Unlike $\gamma_{or,i}$, $\gamma_{in,i}$ is semiempirically estimated with a polynomial equation, determined by fitting the $\gamma_{in,i}$

estimated by the Aerosol Inorganic–Organic Mixtures Functional groups Activity Coefficient (AIOMFAC) (Zuend et al.,

2011) as:

$$\gamma_{in,i} = e^{0.035MW_i - 2.704\ln(O:C_i) - 1.121HB_i - 0.33FS - 0.022(RH)}, \tag{3}$$

where RH and FS is relative humidity (%) and fractional sulfate. Fractional sulfate (FS) is the concentration ratio of total

sulfate to the sum of total sulfate and ammonium ions in aerosol (FS = [SO₄²⁻]/([SO₄²⁻]+[NH₄⁺])) (Zhou et al., 2019). FS,

introduced to determine aerosol acidity, ranges from 0.33 to 1 for ammonium sulfate to sulfuric acid, respectively.

The gas–organic partitioning is governed by Raoult's law, assumed that the saturation vapor pressure of the species is

dependent on the mole fraction of the species in the solution. To consider the oligomerization of organic species in total

concentration ($C_{T,i} = C_{g,i} + C_{or,i} + C_{in,i}$), OM$_P$ is recalculated after OM$_{AR}$ integration with the partitioning model (Schell et

al., 2001), which is reconstructed by including OM$_{AR}$ (Cao and Jang, 2010). OM$_P$ is calculated by the Newton Raphson method

(Press et al., 1992) from $C_{T,i}$ using a mass balance equation:

$$OM_P = \sum_i \left[ C_{T,i} - OM_{AR,i} - C_{g,i}^* \frac{\left(\frac{C_{or,i}}{MW_i}\right)}{\sum_i\left(\frac{C_{or,i}}{MW_i} + \frac{OM_{AR,i}}{MW_{oli,i}}\right) + OM_0} \right], \tag{4}$$

where $OM_0$ (mol m⁻³), $C_{g,i}^* (= \frac{1}{K_{or,i}})$, and $MW_{oli,i}$ (g mol⁻¹) are the concentration of pre-existing OM, the effective saturation

concentration of $i$, and the molecular weight of the dimer ($i$), respectively.

**3.3 SOA formation via aerosol phase reaction**

Organic matter (OM$_{AR}$) is generated via aerosol phase reaction in both organic and inorganic phases, as well as gas-particle

partitioning (OM$_P$). OM$_{AR}$ is estimated as a second order reaction product from condensed organics based on the assumption

of a self-dimerization reaction of organic compounds in media (Im et al., 2014;Zhou et al., 2019;Odian, 2004):

$$\frac{dC'_{or,i}}{dt} = -k_{o,i}C'^2_{or,i}, \tag{5}$$

$$\frac{dC'_{in,i}}{dt} = -k_{AC,i}C'^2_{in,i}, \tag{6}$$





where $C'_{or,i}$ and $C'_{in,i}$ are the concentration of $i$ in the organic and inorganic aerosol phase (mol L$^{-1}$), respectively. The reaction rate constant in the aqueous phase ($k_{AC,i}$) and organic phase ($k_{o,i}$) are determined:

$$k_{AC,i} = 10^{0.25pK_{BH_i^+}+1.0X+0.95R_i+\log(a_w[H^+])-2.58}, \tag{7}$$

$$k_{o,i} = 10^{0.25pK_{BH_i^+}+0.95R_i+1.2\left(1-\frac{1}{1+e^{0.05(300-MW_{or})}}\right)+\frac{2.2}{1+e^{6.0(0.75-O:C)}}-10.07}, \tag{8}$$

where $k_{AC,i}$ (L mol$^{-1}$ s$^{-1}$) is the second order rate constant which is a rate determining step for polymerization to form polyacetal

(Jang et al., 2005;Jang et al., 2006). $k_{AC,i}$ is semiempirically determined from $R_i$, the protonation equilibrium constant ($pK_{BH_i^+}$), excess acidity (X) (Cox and Yates, 1979;Jang et al., 2006), water activity ($a_w$), and the proton concentration [H$^+$] (Im et al., 2014;Zhou et al., 2019). $k_{o,i}$ is determined by extrapolating $k_{AC,i}$ to the neutral condition in the absence of salted aqueous solution to process oligomerization in organic phase. $k_{o,i}$ is calculated without X, $a_w$, and [H$^+$] terms because $a_w$, [H$^+$], and X converged to zero in the absence of wet inorganic seed.

**3.4 Integration of UNIPAR with SAPRC**

In this study, the UNIPAR model was coupled with SAPRC07TC (UNIPAR-SAPRC) to yield the concentration of RO$_2$, HO$_2$, and HC as a function of time under the given conditions. RO$_2$ and HO$_2$ concentrations, and the consumptions of biogenic HCs by four different oxidation pathways were applied to the predetermined mathematical equations, established from explicit gas mechanism to estimate the concentration of lumping species produced from each oxidation pathways. The resulting

concentration of lumping species are plugged into the UNIPAR model to simulate the SOA formation via the gas-particle partitioning and aerosol phase reaction. The reaction rate constant of β-caryophyllene in SAPRC07TC was adjusted based on that from the MCM mechanism (Jenkin et al., 2012). The resulting gas mechanism of biogenic HCs in SAPRC07TC was summarized in the Table S1.

For the SOA simulation with NO$_3$ radicals in the presence of wet inorganic seed aerosol, the heterogeneous hydrolysis of N$_2$O$_5$

was included in gas mechanisms. N$_2$O$_5$ forms via the equilibrium reaction of a NO$_3$ radical and NO$_2$ in gas phase but rapidly hydrolyzed by the interfacial process on the surface of salted aqueous aerosol to form nitric acid (Galib and Limmer, 2021).

**4 Results and Discussions**

**4.1 Simulation of chamber data with the UNIPAR model**

The predictability of the UNIPAR-SAPRC model was demonstrated by simulating SOA data obtained from the oxidation of

three biogenic HCs (isoprene, α-pinene, and β-caryophyllene) in the UF-APHOR chamber under the various environmental conditions, such as NO$_x$ level, inorganic seed conditions, and temperature in both day and night (Table 1).

Figure 2 shows the simulated total SOA mass (OM$_T$, solid line) and OM$_P$ (dotted line) by the UNIPAR-SAPRC. The predicted SOA mass approached with four oxidation paths well accords with the observed SOA mass (symbol). For the ozonolysis of all three biogenic HCs of this study, OM$_P$ attributes more to SOA formation in the presence of NO$_x$ (Fig. 2 (b), (c), (e), (f), (h),

and (j)) than in the absence of NO$_x$ (Fig. 2 (a), (d), (g), and (i)). This suggests the importance of NO$_3$ radicals at nighttime. The SOA formation increased with wet inorganic seed due to aqueous phase reactions of reactive organic species, rendering the reduction of OM$_P$ as seen in Fig. 2 (e and f). In the same manner, SOA formation with acidic seed increased but the fraction of OM$_P$ of total SOA mass decreased (Fig. 2 (g)). However, in the presence of aqueous salts, NO$_3$ radicals rapidly react with NO$_2$ to form dinitrogen pentoxide (N$_2$O$_5$), which can undergo heterogeneous hydrolysis reaction on the surface of wet aerosol

particles to form nitric acid (HNO$_3$) (Brown et al., 2006;Hu and Abbatt, 1997;Galib and Limmer, 2021). Thus, the seed effects



observed in the presence of NO$_x$ from the chamber-generated SOA was mainly attributed to the seed effect on the SOA formation form the ozonolysis products.

Figure 3 illustrates the chamber-generated SOA mass (symbol) under ambient sunlight, the simulated OM$_T$ (solid line), and OM$_P$ (dotted lines). The daytime simulation approached by the four oxidation paths with the UNIPAR-SAPRC model also well agreed with the SOA mass generated under various experimental conditions. For both isoprene SOA (Fig.3 (a) vs. (d)) and β-caryophyllene SOA (Fig. 3 (c) vs. (f)), a clear NO$_x$ effect appeared for those performed under similar experimental conditions measurement and simulation during daytime, showing the higher SOA mass with the greater HC/NO$_x$ level (lower NO$_x$) as previously reported in many studies (Carlton et al., 2009;Tasoglou and Pandis, 2015). The impact of acidic seed on α-pinene SOA formation (Fig. 3 (b) and (e)) was also simulated with the model as reported in other studies (Yu et al., 2021;Han et al., 2016;Kristensen et al., 2014). However, β-caryophyllene SOA was relatively insensitive to the aerosol acidity, which disagreed with the previous observations (Chan et al., 2011;Offenberg et al., 2009). The SOA yield from β-caryophyllene oxidation is very high, even in the absence of salted aqueous phase. Thus, the impact of aqueous reactions on β-caryophyllene can be less dramatic than that of α-pinene. In addition, the large molecules originating from β-caryophyllene oxidation might have a poor solubility in aqueous phase, weakening the impact of aerosol acidity on OM$_{AR}$.

**4.2 Evaluation of Biogenic SOA Potential from Major Oxidation Paths**

Figure 4 displays the potential SOA yield simulated by the UNIPAR-SAPRC model from each oxidation path, which is calculated with the same amount of HC consumption at two different NO$_x$ levels ((a) high NO$_x$: HC/NO$_x$ = 3 ppbC/ppb and (b) low NO$_x$: HC/NO$_x$ = 10 ppbC/ppb). For this calculation, the consumptions of biogenic HCs are set to 50 ppb (138 μg m$^{-3}$), 30 ppb (162 μg m$^{-3}$), and 20 ppb (167 μg m$^{-3}$) for isoprene, α-pinene, and β-caryophyllene, respectively. The temperature is set to 298 K at two different relative humidity (RH) levels (45% and 80%) and three different seed conditions (no seed, wet-AS, and wet ammonium bisulfate (AHS)) with 10 μg m$^{-3}$ of OM$_0$. For the inorganic seeded simulation, the seed concentration is 20 μg m$^{-3}$ (dry mass). Overall, biogenic SOA formation from the O($^3$P) reaction path is absence in daytime.

In this study, the isoprene SOA in the presence of salted aqueous solution is excluded because the simulation focused on LLPS. Isoprene products are very polar and can be mixed with salted aqueous phase (Beardsley and Jang, 2016). Figure 4 shows that the efficient pathways to form isoprene SOA are the NO$_3$- and OH-initiated oxidation in both day and night. The nighttime SOA yield simulated with NO$_3$ radicals at the given condition of Fig. 4 is 12-17%. The contribution of NO$_3$ on SOA yields in daytime can possibly be minimal owing to the rapid photolysis of NO$_3$ (e.g., lifetime = 5s) (Magnotta and Johnston, 1980). The tendency of each oxidation path for isoprene SOA formation shown in Fig. 4 accords with the previous studies (Carlton et al., 2009;Kleindienst et al., 2007;Czoschke et al., 2003) in that the reaction with the OH radical mainly attributes to isoprene SOA. But the reaction with O$_3$ is a minor. Evidently, the SOA yield formed via ozonolysis in the absence of an OH-scavenger was greater than that in the presence of the scavenger in the laboratory work (Kleindienst et al., 2007), suggesting that a sizable fraction of the isoprene aerosol is produced via the OH-oxidation path.

The α-pinene SOA yields is high with ozonolysis and NO$_3$-initiated oxidation in both daytime and nighttime, due to the formation of lowly volatile products from the autoxidation of ozonolysis products (Roldin et al., 2019;Crounse et al., 2013;Bianchi et al., 2019). In general, the photodegradation of oxidation products lessens SOA yields in daytime. As seen in Fig. 4, the NO$_3$-initiated reaction of α-pinene produces a considerable amount of SOA under darkness. The addition of NO$_3$ to the alkene double bond of α-pinene is followed by the addition of an oxygen molecule to form an alkylperoxy radical that can lead to low-volatile peroxide accretion products (ROOR) (Hasan et al., 2021;Bates et al., 2022).

Unlike the diurnal pattern in isoprene and α-pinene SOA yields as shown in Fig. 4, that of β-caryophyllene SOA yield shows higher in daytime than in nighttime. Similar to isoprene and α-pinene products, β-caryophyllene products can be degraded via





photolysis and shorten their carbon number. However, the molecular weight of β-caryophyllene oxidation products is generally much higher than that in isoprene or α-pinene products. Even after photodegradation of β-caryophyllene products, the product volatility is still low enough to significantly partition to the aerosol phase and heterogeneously form SOA. Under darkness, the β-caryophyllene SOA formation potential is the highest with ozonolysis, followed by the $NO_3$-initiated oxidation. At given

simulation conditions in Fig. 4, the β-caryophyllene SOA yield ranges from 26% to 35% for the OH-initiated oxidation and 21-32% for ozonolysis. The SOA yields from the reaction of β-caryophyllene with OH radicals and $O_3$ are found in other laboratory studies (i.e., SOA yields with OH: 17-68%, those with $O_3$: 5-46%) (Chan et al., 2011;Jaoui et al., 2013;Tasoglou and Pandis, 2015).

Figure 4 also shows the SOA potentials at the two different $NO_x$ levels for each oxidation path. Overall, the $NO_3$-initiated SOA

yield drops in decreasing the $NO_x$ level, because the $RO_2$ that forms via the reaction of biogenic HC with $NO_3$ radical followed by the addition of an oxygen molecule can react with $HO_2$ radicals to form organic hydroperoxide, which yields little SOA mass (Bates et al., 2022;Ng et al., 2008). At the low $NO_x$ level, the oxidation of HCs with the OH radical tends to form a large amount of SOA mass. For α-pinene and β-caryophyllene, the low $NO_x$ level increases reactive organic products and thus SOA grows rapidly via heterogeneous reactions. The OH-initiated isoprene SOA yields increases with reducing the $NO_x$ level

because of the formation of epoxy-diol at the low $NO_x$ levels (Kroll et al., 2006). The SOA yields from ozonolysis of α-pinene decreases by increasing the $NO_x$ level. The autoxidation path of the α-pinene ozonolysis product can be suppressed under the high $NO_x$ level (Bianchi et al., 2019). Thus, it lowers the formation of low volatility products. For β-caryophyllene, the nighttime SOA formation from ozonolysis increases with reducing the $NO_x$ level, because the internal rearrangement of ozonolysis products to form the secondary ozonide competes with the reaction of these ozonolysis products with NO or $NO_2$

(Jenkin et al., 2012). The further oxidation of the secondary ozonide products yield low-volatile products. However, the ozonolysis β-caryophyllene SOA under sunlight increases with increasing the $NO_x$ level. The first generation of the ozonolysis of β-caryophyllene forms product containing an alkene double bond and an aldehyde group, which can react with a $NO_3$ radical to form high-carbon peroxy radicals. Furthermore, these peroxy radicals can produce a variety of organic products that form SOA via heterogeneous reactions and partitioning of the low volatility products (Jenkin et al., 2012;Li et al., 2011).

Regardless of HCs and oxidation pathways, the impact of neutral seed (wet-AS) on biogenic SOA formation is insignificant. The impact of the acidic seed (wet-AHS) on α-pinene SOA formation is various depending upon the oxidation path. For daytime SOA formation, the significant impact of acidic seed on SOA formation is observed as previously reported (Yu et al., 2021;Han et al., 2016). For nighttime, no significant impact of acid-catalyzed reactions on the α-pinene SOA originating from the $NO_3$-initiated pathway appears, because the SOA forms from lowly volatile ROOR products that are insensitive to aerosol

acidity (Boyd et al., 2017). For β-caryophyllene, overall, the increase in SOA yields due to aqueous phase reactions are not significant for all oxidation pathways. Only a small increase in the β-caryophyllene SOA yield in the presence of acidic seed (wet-AHS) appears for the $O_3$-initiated oxidation path at nighttime and daytime.

The chamber-generated SOA mass is influenced by the deposition of organic vapor to the chamber wall. The simulation of SOA yields in Fig. 4 is performed with the model parameters obtained in the presence of the chamber wall. To investigate the

SOA formation in the ambient air, the wall-free SOA model parameter has recently been derived by (Han and Jang, 2022). Figure S3 illustrates the SOA yields predicted in the absence of the deposition of organic vapor to the chamber wall. Overall, the effect of acidic seed on SOA formation is reduced by the correction of model parameters for the wall artifact. α-Pinene SOA is more influenced by gas-wall partitioning than isoprene or β-caryophyllene SOA, especially for the OH radical and $NO_3$ radical oxidation paths.





**4.3 Sensitivity of Biogenic SOA Formation to Major Variables**

Figure 5 illustrates the simulated SOA yield from (a) isoprene, (b) α-pinene, and (c) β-caryophyllene in both daytime (solid line) and nighttime (dashed line) under the various temperature, ranging from 278K to 308K, at the given reference condition. The bias from gas-wall partitioning is corrected in this simulation by the amended model parameter (Han and Jang, 2022). For the daytime, the SOA formation is simulated under the reference sunlight intensity (Fig. S1), measured on 06/19/2015 at the UF-APHOR. The simulation is performed for the urban atmosphere where the $NO_x$ level is high (HC/$NO_x$ = 3 ppbC/ppb), because the concentration of $O_3$ and $NO_3$ radicals are relatively high in the polluted atmosphere. The SOA formation is simulated with 10 μg m$^{-3}$ of $OM_0$ at the constant RH (50%) with the fixed initial concentration of isoprene, α-pinene, and β-caryophyllene as 50 ppb, 30 ppb, and 24 ppb, respectively. The sensitivity of SOA mass to temperature is simulated for α-pinene and β-caryophyllene under NS and wet-AHS (20 μg m$^{-3}$) conditions. The contribution of each oxidation pathway to the consumption of biogenic HC is illustrated in Fig. S2 (a).

Figure 5 shows that isoprene and α-pinene produce more SOA mass at nighttime than daytime. In Fig. S2 (a), more than 90% of isoprene and α-pinene are consumed by $NO_3$ radicals and $O_3$ at night, while they are mainly consumed by OH radicals in daytime. As discussed in Sect. 4.2, their oxidation paths with $NO_3$ radicals or $O_3$ at night produce considerably high SOA mass yields. For β-caryophyllene, the majority of HC is consumed by $O_3$ in both day and night. In addition, OH radicals' contribution to the HC consumption is positively correlated to the ozonolysis, which produces the OH radicals as a byproduct. The daytime SOA yield (Fig. S3) from the β-caryophyllene ozonolysis and OH-initiated oxidation are similar or greater than that in night due to the formation of reactive products for oligomerization during multi-generation photochemical oxidation as discussed in section 4.2.

In the absence of the inorganic seed, the nighttime SOA from all three HCs is more sensitive to the temperature than that produced in daytime. At nighttime, the biogenic HCs, primarily consumed by $O_3$ and $NO_3$ radicals, produce the semi-volatiles that form SOA dominantly by the partitioning process. The products formed via the photooxidation in daytime are multi-functional and reactive for aerosol phase reactions, and thus less sensitive to temperature.

The daytime α-pinene SOA formation is enhanced by the acid-catalyzed reaction up to 1.2 times (Fig. 5 (b)). However, the nighttime α-pinene SOA mass decreases by introducing inorganic seeds due to the decay of the $NO_3$ radicals through the heterogeneous hydrolysis of $N_2O_5$, which thermodynamically forms $NO_3$ radicals. The reduction of $NO_3$ radicals results in less contribution of the $NO_3$ path that can lead a high yield SOA formation (Fig. S3 (a)). In both nighttime and daytime, β-caryophyllene SOA mass increases up to 1.1 times by aqueous phase reactions. The impact of wet seed on isoprene SOA is not discussed in this study because the mixing state of isoprene products and wet-salt aerosol is not governed by LLPS. Studies have shown that the impact of aqueous reaction isoprene SOA is greater than α-pinene (Beardsley and Jang, 2016;Carlton et al., 2009).

Figure 6 illustrates the sensitivity of SOA yields to the $NO_x$ level at a given reference condition with 10 μg m$^{-3}$ of $OM_0$. The temperature and RH are set to 298 K and 50%, respectively. The simulations are performed in the absence of the gas-wall partitioning (Han and Jang, 2022) with the given initial concentration of isoprene, α-pinene, and β-caryophyllene as 50 ppb, 30 ppb, and 24 ppb, respectively. The SOA simulations from α-pinene and β-caryophyllene are performed under NS and wet-AHS conditions. The dry mass of wet-AHS is set to 20 μg m$^{-3}$. Figure 6 shows the impact of $NO_x$ to ozonolysis SOA at nighttime. Despite of a large increase in α-pinene SOA potential due to the gas-wall loss correction (Fig. 4 vs. Fig. S3), the highest yield in daytime SOA appears with β-caryophyllene, followed by α-pinene and isoprene. In nighttime, the simulation suggests that α-pinene SOA yield could be higher than the β-caryophyllene SOA at the high $NO_x$ zone due to the high SOA potential from the $NO_3$-initiated oxidation path.



For α-pinene and isoprene, daytime SOA yields gradually decrease with increasing $NO_x$, but nighttime SOA yields drastically increase in the high $NO_x$ region. In daytime, the high $NO_x$ level increases the reactions of NO with $RO_2$, leading to relatively volatile gas products lowering SOA yields (Yu et al., 2021;Carlton et al., 2009;Hallquist et al., 2009). At night, as seen in Fig. S2, the high-yield $NO_3$-initiated path contributes more with the high $NO_x$ level. The $NO_x$ effects on nighttime biogenic SOA formation in the presence of inorganic seed are lesser than those in daytime due to the removal process of $NO_3$ radicals via the heterogeneous hydrolysis of $N_2O_5$ on the wet inorganic seed. For β-caryophyllene, the $NO_x$ effects on SOA yield shows the opposite trend to α-pinene and isoprene SOA. The daytime β-caryophyllene SOA yields from ozonolysis decreases with reducing $NO_x$ level. Figure S2 suggests that the β-caryophyllene is mainly consumed by the $O_3$ initiated path, and thus the β-caryophyllene SOA yield in daytime changes by $NO_x$ level (Fig. S3) due to the different product distributions from the ozonolysis as discussed in section 4.2.

## 4.4 Nocturnal Biogenic SOA Formation in the Presence of gasoline

To investigate the influence of anthropogenic HC on the terpene SOA formation at night, 75 ppb α-pinene is oxidized with 120 ppb $O_3$ in the presence of 3000 ppbC gasoline fuel. The details of the experimental conditions are summarized in Table 2. Fig.7 illustrates the UNIPAR-SAPRC simulated SOA mass ($OM_T$, solid line) and the observed chamber-generated SOA mass (symbol). The aromatic HCs in gasoline fuel can be oxidized with the OH radicals, which is produced as a by-product from the ozonolysis of α-pinene (Finlayson-Pitts and Pitts Jr, 1999). The simulation suggests that the SOA formation in the α-pinene and gasoline cocktail mainly originates from α-pinene oxidation products. The α-pinene SOA (red line) contributes from 95 to 98% of total SOA in the absence of inorganic seed (Fig. 7 (a)), and it slightly decreases to 93-94% in the presence of wet-AS because the gasoline aromatic oxidation products are highly reactive in the aqueous phase (Han and Jang, 2022). The simulated α-pinene SOA mass in the presence of gasoline fuel is also compared to that in the absence of gasoline in Fig. 7. Interestingly, the SOA formation in the presence of gasoline is elevated by 30% compared to that in absence. Under the ozone excess condition of this study, the oxidation of α-pinene is mainly dominated by ozonolysis in the presence of gasoline, because gasoline competes with α-pinene to react with OH radicals. As seen in Fig. 4, α-pinene ozonolysis is capable of yielding higher SOA mass than the α-pinene OH reaction.

The impact of anthropogenic hydrocarbons (gasoline) on α-pinene SOA is also demonstrated for different $NO_x$ levels and seed conditions without the chamber wall bias in Fig. S4. In the absence of inorganic seed, the simulation shows the higher SOA yields with the greater $NO_3$ contribution at the higher $NO_x$. In the presence of inorganic seed, the contribution of $NO_3$ on SOA formation decreases due to the heterogeneous hydrolysis of $N_2O_5$, which forms the $NO_3$ radicals as discussed in section 4.3. The effects of gasoline to total SOA formation decrease with increasing the $NO_x$ level, because less ozonolysis results in less production of OH radicals. In addition, the aromatic SOA yield is generally smaller at the higher $NO_x$ level (Han and Jang, 2022).

## 4.5 Impact of Model Parameters on SOA mass

The uncertainty test of SOA mass is performed for two major processes associated with partitioning ($P_L$) and aerosol phase reactions in both the organic phase and the aqueous phase ($k_o$, and $k_{AC}$) in the absence of chamber wall bias. The uncertainty in SOA mass in Fig. S5 is performed by increasing/decreasing $P_L$, $k_o$, and $k_{AC}$ as a factor of 1.5/0.5, at the high $NO_x$ level (HC/$NO_x$ = 3 ppbC/ppb) with 10 μg m$^{-3}$ of $OM_0$. The daytime SOA mass is simulated with the sunlight profile on 06/19/2015 near summer solstice (Fig. S1). Temperature and RH are set as 298K and 40%, respectively. The amount of both wet-AS and wet-AHS is fixed to 20 μg m$^{-3}$ (dry mass). In the model, $P_L$ was determined based on the group contribution (Stein and Brown, 1994) with the reported error as a factor of 1.45 (Zhao et al., 1999). $k_{AC}$ was semi-empirically determined by correlating model compound data with the [$H^+$] and liquid water contents, and $R_i$ (Eq. 7). $k_o$ was obtained by extending the $k_{AC}$ calculation to



the neutral condition in the absence of salted aqueous solution to process oligomerization in organic phase by eliminating X, $a_w$, and [H$^+$] terms (Eq. 8). Among the reaction systems in Fig. S5, α-pinene daytime SOA formation is the most responsive to the change of the three model parameters. $P_L$ is more influential on all three biogenic SOA formation than $k_o$, and $k_{AC}$ at the given simulation condition.

## 5 Atmospheric Implication

In this study, the biogenic SOA produced from the reaction of isoprene, α-pinene, or β-caryophyllene with four major atmospheric oxidants (OH radicals, O$_3$, NO$_3$ radicals, and O($^3$P)) was simulated with the UNIPAR-SAPRC model and applied to interpretation of their diurnal pattern. The simulation (Fig. 6) indicated that isoprene and α-pinene SOA yields in daytime increased by decreasing the NO$_x$ level, but they showed the opposite tendency at night. This trend accords with the previous laboratory studies and field observations (Rollins et al., 2012;Yu et al., 2021;Carlton et al., 2009;Hallquist et al., 2009;Fry et

al., 2018). As seen in Fig. S2, the NO$_3$ radical significantly contributed to the biogenic HC consumption at night, although its contribution can be lesser in the presence of wet inorganic aerosol. Field observations have shown a considerable contribution of NO$_3$ radicals to biogenic HC oxidation at night, up to 58% of the total oxidation paths (Ng et al., 2017;Edwards et al., 2017).

Owing to the efforts of the governmental agency, the NO$_x$ emission from anthropogenic sources has gradually decreased, and it impacts the NO$_3$ concentrations. For example, the nighttime oxidation path in the southeast US is in transition from NO$_x$-

dominance to O$_3$-dominance (Edwards et al., 2017), due to the reduction of the NO$_x$ emission (Russell et al., 2012). The fate of SOA formation caused by the nocturnal chemistry under the reduction of the NO$_x$ emission is, however, complex due to several reasons. Under the urban set, the biogenic HCs are oxidized in the presence of the complex cocktail of anthropogenic pollutants (i.e., aromatic HCs, SO$_2$ and NO$_x$). As discussed in section 4.4, the reduction of NO$_x$ can lessen biogenic SOA mass at night (Figure 4 and Figure 6), although it increases aromatic SOA originating from the oxidation with OH radical. On the

other hand, the reduction of NO$_x$ can increase daytime biogenic SOA burdens in urban air. Additionally, NO$_2$ can react with OH radicals at high NO$_x$ zones to form HNO$_3$, which is semi-volatile and can condense onto the preexisting particles at the low temperature (Wang et al., 2020). Under the rural environment where the NO$_x$ level is low, the reduction of NO$_x$ can generally increase biogenic SOA formation in both daytime and nighttime, but its impact could be trivial compared to that in the high NO$_x$ zone.

Electrolytic inorganic salts are ubiquitous in the urban air because sulfate and nitrate are produced by the atmospheric oxidation of SO$_2$ and NO$_x$, respectively (Finlayson-Pitts and Pitts Jr, 1999). For last two decades, numerous studies have shown the significant increase in biogenic SOA mass in the presence of acidic seed due to the acid-catalyzed heterogeneous reactions (Jang et al., 2002;Czoschke et al., 2003;Offenberg et al., 2009;Surratt et al., 2010;Beardsley and Jang, 2016;Hallquist et al., 2009). Most previous studies have demonstrated the impact of acidic aerosol on daytime SOA. In this study, both simulations

and chamber observations (Figure 4) showed a weak seed impact on ozonolysis biogenic SOA. The nighttime biogenic SOA formation via the NO$_3$-initiated oxidation path was even less sensitive to the seed condition compared to ozonolysis SOA. Boyd et al. (2017) also reported a similar observation for monoterpenes (Boyd et al., 2017). In addition, the contribution of NO$_3$ radicals on nighttime SOA formation under the high NO$_x$ zone can be less in the presence of salted aqueous solution. The heterogeneous hydrolysis of N$_2$O$_5$ (Brown et al., 2006;Hu and Abbatt, 1997) on the surface of inorganic seed lessens the

concentration of NO$_3$ radicals. However, when nighttime SOA formation is dominated by ozonolysis at the low NO$_x$ zone, the SOA formation can be enhanced by wet seed (Fig. S3).

The emission of biogenic HCs is regulated by temperature and light intensity (Holzke et al., 2006;Chen et al., 2020;Petron et al., 2001;Goldstein et al., 1998). Thus, there is a diurnal pattern in the biogenic HC emission, showing the higher biogenic HC emission during daytime (Holzke et al., 2006). The concentrations of O$_3$ and NO$_2$ are generally high at daytime due to the



oxidation of hydrocarbon, involving the photochemical cycle of $NO_x$ and the anthropogenic emission from human activities. The emission of biogenic HCs is lower by 3-4 times at night (Holzke et al., 2006) than that in daytime considering emission rate and the mixing height, but biogenic SOA yields significantly increase at night because of different oxidation paths and temperature reduction. For example, terpene and isoprene SOA yields increase almost by one order of magnitude as discussed in Fig. 5.

The model uncertainties to predict SOA mass mainly originates from gas mechanisms and aerosol phase reactions. The model approach by using explicit mechanisms is complex and time demanding, but it can improve a predictability of multiphase partitioning of products and heterogeneous reactions under varying environmental conditions. For example, the daytime β-caryophyllene SOA of this study was underpredicted as seen in Fig. 3, suggesting that the improvement of explicit gas mechanisms is essential to better predict SOA formation. In the model, the multiphase reaction of biogenic HC is individually

treated with four different oxidation paths. Neither complex cross reactions between $RO_2$ radicals nor the long-term aging process of multiple generation products were not fully considered, which can be a source of the bias in SOA prediction. In the presence of inorganic seed, heterogenous hydrolysis of $N_2O_5$ was assumed to be very rapid. However, the variation of aerosol constituents can influence the accommodation coefficient of $N_2O_5$. For example, the heterogeneous hydrolysis of $N_2O_5$ on organic-coated aerosol can be slower than that in salted aqueous phase (Anttila et al., 2006). To accurately model the reaction

of biogenic HCs with $NO_3$ radicals, the impact of aerosol compositions on the heterogeneous reaction of $N_2O_5$ on the surface of aerosol needs to be investigated in future. In addition, the hydrolysis of organonitrates in aerosol phase and the aging of particle organic matter were not included in the model.

*Code availability.* Code to run the SOA model in this study is available upon request.


*Data availability.* The chamber data and simulated results used in this study are available upon request.

*Author contribution.* MJ and SH designed the experiments, and SH carried them out. SH prepared the manuscript with contributions from MJ.


*Competing interest.* The authors declare that neither they nor their co-author has any conflict of interest.

*Acknowledgments.* This research was supported by the National Institute of Environmental Research (NIER2020-01-01-010); the National Science Foundation (AGS1923651); and the Fine Particle Research Initiative in East Asia Considering National

Differences (FRIEND) Project through the National Research Foundation of Korea (NRF) funded by the Ministry of Science and ICT (2020M3G1A1114556).





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





**Tables and Figures**

Table 1. Experimental conditions for the oxidation of biogenic HCs in the UF-APHOR chamber.

| Exp. ID | Date | Initial condition | | | | | Temp (K) | %RH | max OM ($\mu g\ m^{-3}$) | Light condition[4] | Figures |
| | | HC (ppbC) | HC/NOx (ppbC/ppb) | Seed[1] | Seed mass[2] ($\mu g\ m^{-3}$) | $OM_0$[3] ($\mu g\ m^{-3}$) | | | | | |
| Isoprene ($C_5H_8$) | | | | | | | | | | | |
| IS01 | 10/04/2021 | 750 | - | - | | 5 | 295-302 | 44-81 | 27 | N | Fig. 2 (a) |
| IS02 | 10/07/2021 | 782 | 13.3 | - | | 5 | 297-301 | 42-56 | 31 | N | Fig. 2 (b) |
| IS03 | 10/20/2021 | 750 | 3.9 | - | | 5 | 292-298 | 36-75 | 30 | N | Fig. 2 (c) |
| IS04 | 12/16/2021 | 696 | 5.6 | - | | 5 | 291-310 | 16-38 | 116 | Y | Fig. 3 (a) |
| IS05 | 01/27/2015 | 839 | 17.4 | - | | 3 | 279-298 | 27-66 | 62 | Y | Fig. 3 (d) |
| α-pinene ($C_{10}H_{16}$) | | | | | | | | | | | |
| AP01 | 03/19/2021 | 84 | - | - | | 4 | 282-306 | 42-95 | 157 | N | Fig. 2 (d) |
| AP02 | 06/23/2021 | 92 | - | SA | 100 | 4 | 296-899 | 72-88 | 96 | N | Fig. 2 (e) |
| AP03 | 06/23/2021 | 79 | - | wAS | 100 | 4 | 296-300 | 89-100 | 80 | N | Fig. 2 (e) |
| AP04 | 09/09/2021 | 64 | 10.5 | - | | 5 | 296-299 | 34-42 | 37 | N | Fig. 2 (f) |
| AP05 | 09/09/2021 | 58 | 10.5 | SA | 85 | 5 | 297-299 | 41-72 | 27 | N | Fig. 2 (f) |
| AP06 | 09/20/2021 | 61 | 3.7 | - | | 6 | 297-301 | 37-55 | 28 | N | Fig. 2 (g) |
| AP07 | 09/20/2021 | 59 | 4.1 | SA | 87 | 6 | 298-302 | 37-55 | 33 | N | Fig. 2 (g) |
| AP08 | 11/04/2021 | 60 | 2.3 | dAS | 40 | 5 | 288-294 | 32-45 | 58 | N | Fig. 2 (h) |
| AP09 | 11/04/2021 | 60 | 2.3 | - | | 5 | 289-293 | 44-66 | 63 | N | Fig. 2 (h) |
| AP10 | 08/28/2019 | 124 | 11.3 | - | | 4 | 296-320 | 14-40 | 23 | Y | Fig. 3 (b) |
| AP11 | 08/28/2019 | 130 | 10.7 | SA | 50 | 4 | 296-317 | 32-54 | 98 | Y | Fig. 3 (b) |
| AP12 | 07/18/2019 | 142 | 4.9 | SA | 60 | 3 | 294-320 | 13-42 | 52 | Y | Fig. 3 (e) |
| AP13 | 07/18/2019 | 139 | 4.6 | - | | 3 | 294-319 | 19-48 | 28 | Y | Fig. 3 (e) |
| β-caryophyllene ($C_{15}H_{24}$) | | | | | | | | | | | |
| BC01 | 11/10/2021 | 50 | | | | 4 | 292-299 | 29-67 | 95 | N | Fig. 2 (i) |
| BC02 | 11/10/2021 | 50 | | SA | 70 | 4 | 293-298 | 36-72 | 73 | N | Fig. 2 (i) |
| BC03 | 11/23/2021 | 40 | 4.2 | | | 3 | 278-293 | 40-72 | 65 | N | Fig. 2 (j) |
| BC04 | 12/03/2021 | 50 | 10.5 | SA | 120 | 3 | 281-308 | 23-90 | 219 | Y | Fig. 3 (c) |
| BC05 | 12/03/2021 | 50 | 10.5 | | | 4 | 282-308 | 30-95 | 256 | Y | Fig. 3 (c) |
| BC06 | 12/10/2021 | 50 | 3.8 | | | 3 | 287-310 | 25-77 | 100 | Y | Fig. 3 (f) |
| BC07 | 12/10/2021 | 50 | 3.8 | SA | 150 | 3 | 288-311 | 26-72 | 87 | Y | Fig. 3 (f) |

[1]NS, SA, wAS, and dAS indicate non-seeded, sulfuric acid seed, wet ammonium sulfate seed, and dry ammonium sulfate seed, respectively. [2]The seed mass is determined as a dry mass, without water mass. [3]The pre-existing organic matter ($OM_0$) is determined for the chamber air prior to the injection of inorganic seed and HC. [4]Sunlight indicates the light condition of the experiment. N denotes the nighttime experiment conducted with no sunlight, and Y denotes the daytime experiments performed under the sunlight with photolysis.







Table 2. Experimental conditions for the oxidation of α-pinene with gasoline fuel in the UF-APHOR chamber.

| Exp. ID | Date | Initial condition | | | | | | Temp (K) | %RH | max OM ($\mu g\ m^{-3}$) |
| | | α-pinene (ppbC) | Gasoline fuel (ppbC) | Seed[1] | $O_3$ (ppb) | Seed mass[2] ($\mu g\ m^{-3}$) | $OM_0$[3] ($\mu g\ m^{-3}$) | | | |
|---|---|---|---|---|---|---|---|---|---|---|
| APGF01 | 11/30/2021 | 750 | 3000 | - | 120 | | 5 | 279-295 | 29-76 | 140 |
| APGF02 | 11/30/2021 | 782 | 3000 | wAS | 115 | 94 | 5 | 279-296 | 38-89 | 120 |

[1]NS, SA, wAS, and dAS indicate non-seeded, sulfuric acid seed, wet ammonium sulfate seed, and dry ammonium sulfate seed, respectively. [2]The seed mass is determined as a dry mass, without water mass. [3]The pre-existing organic matter ($OM_0$) is determined for the chamber air prior to the injection of inorganic seed and HC.






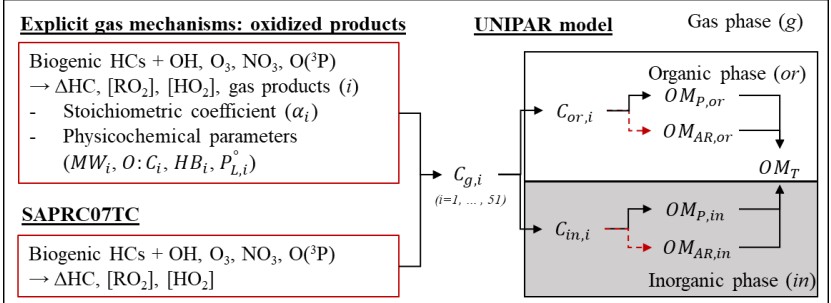

Figure 1. The model structure of the UNIPAR model coupled with SAPRC07TC gas mechanism with model parameters originated from the explicit gas mechanisms. The lumping species and their model parameters were estimated by simulating the explicit gas mechanism and applied to the UNIPAR model simulation. $C_{g,i}$, $C_{or,i}$, and $C_{in,i}$ are the concentration of organic compound ($i$) in gas phase ($g$), organic phase ($or$), and inorganic phase ($in$). $OM_{p,or}$ and $OM_{p,in}$ is the SOA mass generated via gas-particle partitioning in $or$ and $in$, respectively. $OM_{AR,or}$ and $OM_{AR,in}$ is the SOA mass generated via in-particle chemistry in $or$ and $in$, respectively.

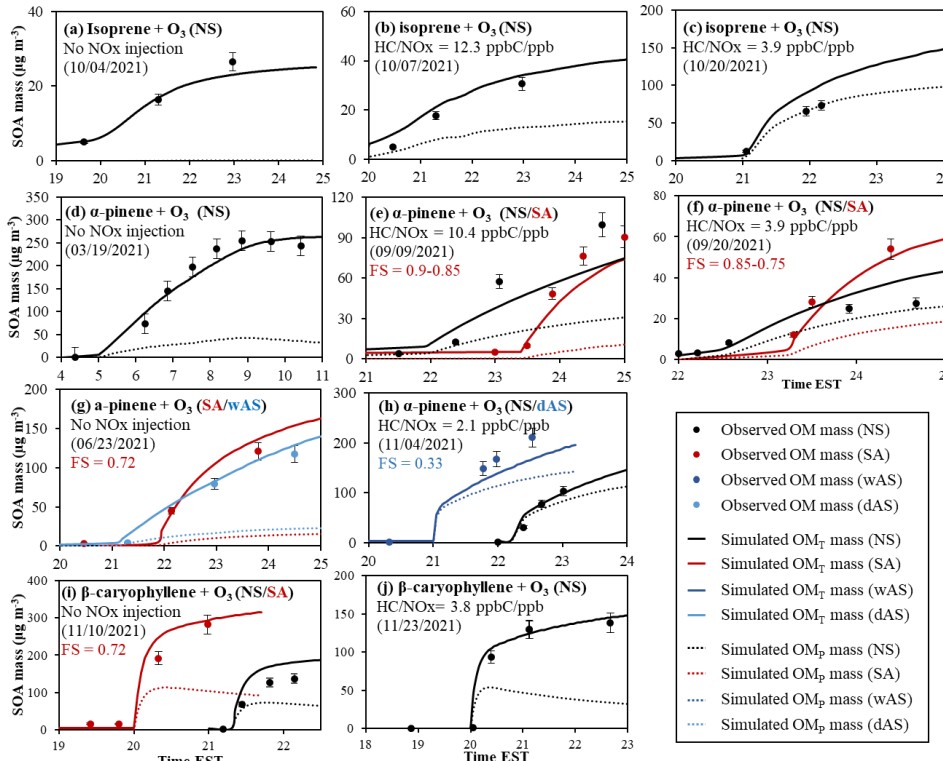

Figure 2. Observed (symbols) and simulated SOA mass (line) for the ozonolysis of isoprene ((a)-(c)), α-pinene ((d)–(h)), and β-caryophyllene ((i) and (j)) at different seed conditions and $NO_x$ levels. SOA mass concentrations are corrected for the particle loss to the chamber wall. The simulated $OM_T$ (solid line) and $OM_{AR}$ (dotted line) are also illustrated. The error (10%) associated with SOA mass was estimated with the instrumental uncertainty.






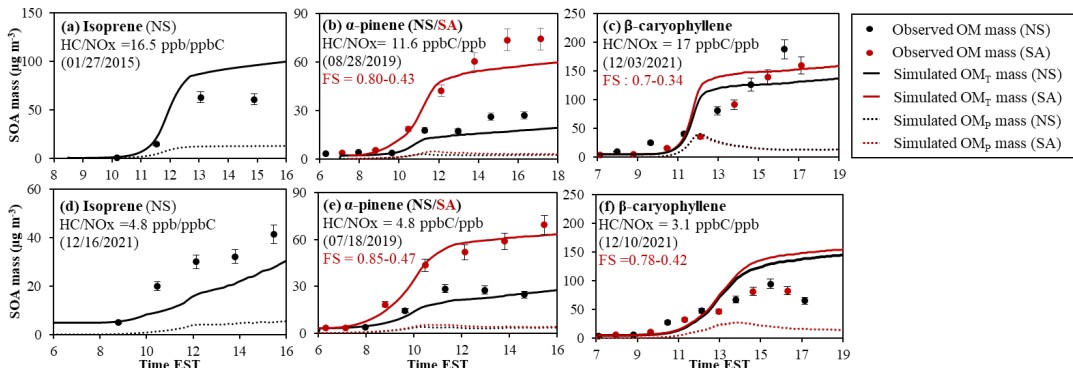

Figure 3. Observed (point) and simulated SOA mass (line) for the photooxidation of isoprene ((a) and (d)), α-pinene ((b) and (e)), and β-caryophyllene ((c) and (f)) at different seed conditions and NO$_x$ levels. SOA mass concentrations are corrected for the particle loss to the chamber wall. The simulated OM$_T$ (solid line) and OM$_{AR}$ (dotted line) are also illustrated. The error (10%) associated with SOA mass was estimated with the instrumental uncertainty.

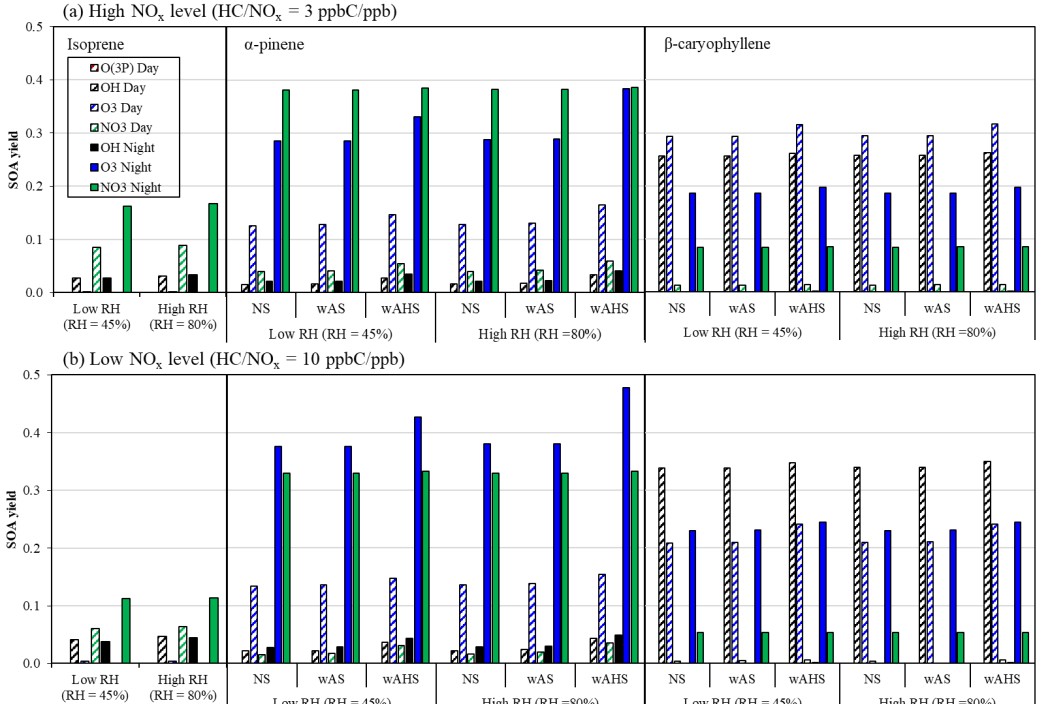

Figure 4. The potential SOA yield from each oxidation path from the given HC consumption under (a) high NO$_x$ level (HC/NO$_x$ = 3 ppbC/ppb) and (b) low NO$_x$ level (HC/NO$_x$ = 10 ppbC/ppb). The consumption of biogenic HC was set to 50 ppb (138 μg m$^{-3}$), 30 ppb (162 μg m$^{-3}$), and 20 ppb (167 μg m$^{-3}$) for isoprene, α-pinene, and β-caryophyllene, respectively. The SOA formation was simulated at 298K under two different RH (45% and 80%) with 10 μg m$^{-3}$ of OM$_0$. For the α-pinene and β-caryophyllene, the SOA formed at three different seed conditions (NS, wAS, wAHS).



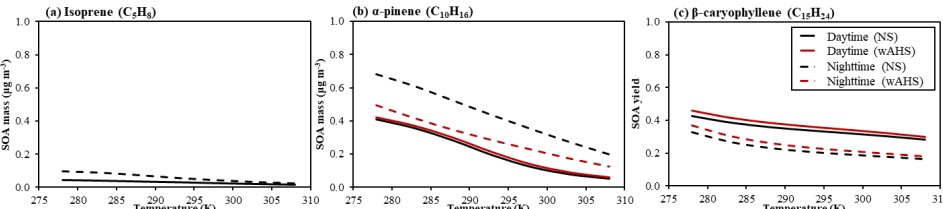

Figure 5. The biogenic SOA yield from (a) isoprene, (b) α-pinene, and (c) β-caryophyllene in both daytime (solid line) and nighttime (dashed line) under the various temperature, ranging from 278K to 308K. The HC/NOₓ level was set to 3 ppbC/ppb. The SOA formation was simulated with 10 μg m$^{-3}$ of OM$_0$ at the 50% of RH with the fixed initial concentration of isoprene, α-pinene, and β-caryophyllene at 50 ppb, 30 ppb, and 24 ppb, respectively. The daytime SOA formation was simulated under the reference sunlight intensity (Fig. S1), which was measured on 06/19/2015 at the UF-APHOR. The wall-free model parameters were applied to simulate SOA formation (Han and Jang, 2022).





725

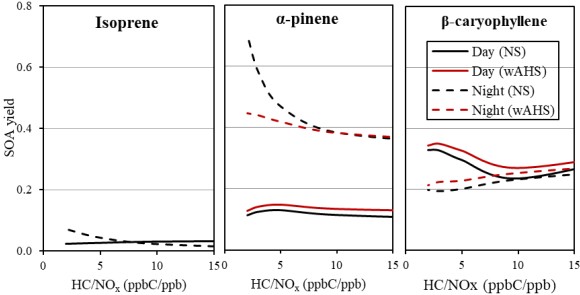

Figure 6. The biogenic SOA yield from (a) isoprene, (b) α-pinene, and (c) β-caryophyllene in both daytime (solid line) and nighttime (dashed line) under the various $NO_x$ levels. The RH and temperature were set to 50% and 298 K, respectively. The SOA formation was simulated with 10 μg m$^{-3}$ of $OM_0$ with the fixed initial concentration of isoprene, α-pinene, and β-caryophyllene as 50 ppb, 30 ppb, and 24 ppb, respectively. The daytime simulation is performed under the reference sunlight

730 intensity (Fig. S1) which was measured on 06/19/2015 at the UF-APHOR. The wall-free model parameters were applied to simulate SOA formation (Han and Jang, 2022).





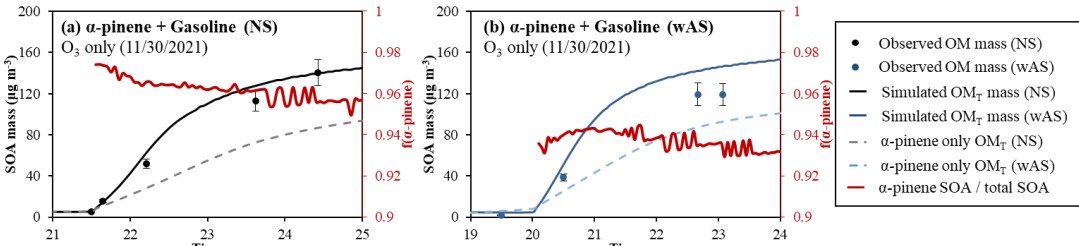

Figure 7. Observed (symbol) and simulated SOA mass (line) for the ozonolysis of α-pinene in the presence of gasoline fuel (a) without and (b) with wet-AS seed. SOA mass concentrations are corrected for the particle loss to the chamber wall. The simulated $OM_T$ (solid line) in the presence of gasoline fuel and that in the absence of gasoline fuel (dashed line) are also illustrated. The dashed lines denote the simulated SOA mass in the absence of gasoline fuel under the same experimental conditions. The fraction of α-pinene SOA to total SOA ($f$(α-pinene)) are illustrated in red lines. The error (10%) associated with SOA mass was estimated with the instrumental uncertainty.