# Peer review of "Modeling Day and Nighttime SOA Formation via Multiphase Reactions of Biogenic Hydrocarbons"

_Atmospheric Chemistry and Physics, 2022_

## Author Comment (AC1)

**Modeling Diurnal Variation of SOA Formation via Multiphase Reactions of Biogenic Hydrocarbons (Manuscript Ref#: acp-2022-327)**

**Response to the Referee1**

We would like to thank the reviewer for their time and thoughtful comments on this manuscript. Their comments are repeated below followed by our response to each comment.

**Overall Comment on the manuscript**

The manuscript "Modeling Diurnal Variation of SOA Formation via Multiphase Reactions of Biogenic Hydrocarbons" presents an attempt to model SOA formation from three different VOC precursors under varies conditions and to validate those results with chamber experiments. To do so the authors have extended their own UNIPAR model with additional oxidation pathways and reactions including product lumping based on volatility. In addition, physical parameters like relative humidity, temperature, etc. from the chamber experiments have been used as conditions for the model simulations. The paper presents some interesting science and an approach that is well worth publication in ACP after major revisions and corrections.

**General remarks:**

a. The authors should try to improve the language of the manuscript to increase its readability and help the reader to understand it more clearly. One common mistake to be found throughout the manuscript is the use of the direct article or its omission in the wrong places. I recommend careful proof reding by a native speaker (of the revised manuscript!).

b. The title suggests investigation of the complete dial cycle. The experiments for individual compounds however do not cover whole day cycles but limited periods (3-12 h total, 2-10h for VOC oxidation/chemistry). In most cases these periods even differ for different compounds/experiments. Therefore, I recommend changing the title to "… under day- and nighttime conditions …" or similar.

c. The abstract is quite long and detailed and partially feels like a conclusion. It might be good to shorten it slightly to give the audience the motivation to read the paper!

d. In several places the authors mention that modeling the gas phase chemistry was done using explicit mechanisms, at other places the talk about "near explicit mechanisms". These expressions are quite ambiguous without clear statements what the authors mean by explicit at the time of use (kinetics, mechanism, et cetera). This is especially true since it is mentioned (in some parts) that the resulting products were lumped based on their volatility, preventing for example explicit modeling of 2nd and higher generation products. I recommend to either rephrase or clarify the extent of this "explicit modelling".

e. Often the authors tend to mix introduction, background, results, and discussion throughout the whole manuscript leading to repetitive statements and elongating it unnecessarily. The manuscript would gain a lot quality wise by clearly separating these different parts (both in term of length and readability). Some parts of the manuscript seem to be more a review paper than an original work. Again, a more in depths presentation and discussion of the results of this work would be very beneficial for the manuscript.

f. Many of the plots and diagrams are very small, and the axes are unfortunately scaled. This makes it really difficult to follow the authors discussion of their results! The graphical presentation of the results should also be improved!

**Response**

a. Both the language of the manuscript and the flow have been improved to help the reader to provide clear description.

b. In order to response to the reviewer, the title of the manuscript has been changed from "Modeling Diurnal Variation of SOA Formation via Multiphase Reactions of Biogenic Hydrocarbons" to "Modeling day and nighttime SOA formation via Multiphase Reactions of Biogenic Hydrocarbons" in the revised manuscript

c. Abstract has been modified to provide better flow in the context and concise.

d. Words "near explicit mechanisms" were replaced with "explicit mechanisms" in the revised manuscript.

e. Both the language of the manuscript and the flow have been improved to help the reader to provide clear description.

f. Figures have been revised with a larger frames and characters.

**Individual comments:**

1. P1 L9: "… intensively evaluated …": I suggest removing "intensively". It is an unnecessary exaggeration.

   **Response:** The manuscript has been revised based on this comment. "intensively" is removed from the manuscript.

2. P1 L13: How can the gas mechanism implement the MCM? Please rephrase!

   **Response:** The manuscript has been revised based on this comment. The gas oxidation process has been simulated by including MCM, PRAM, and formation of low volatile compounds.

   L13: "The gas mechanisms include the Master Chemical Mechanism (MCM v3.3.1), the reactions that formed low volatility products via peroxy radical ($RO_2$) autoxidation, and self- and cross-reactions of nitrate-origin $RO_2$."

3. P1 L15: How did integration with the SAPRC gas mechanism "increase feasibility"? Please clarify or rephrase.

   **Response:** This sentence has been removed from the abstract in the revised manuscript.

4. P1 L21: What are background NOx levels in the simulation (and the chamber experiments)?

   **Response:** The background $NO_x$ level was low as ~3 ppb before experiment. For the experiment, the $NO_x$ levels varied based on the chamber conditions. $HC/NO_x$ levels were classified into two groups, low $NO_x$ ($HC/NO_x$ < 5 ppbC/ppb) and high $NO_x$ ($HC/NO_x$ < 5 ppbC/ppb) as seen in Table 1. The simulation was performed under the same conditions with experiment to predict chamber-generated SOA data. The $HC/NO_x$ ratio was set as 3 ppbC/ppb for high $NO_x$ conditions and 10 ppbC/ppb for low $NO_x$ conditions for the sensitivity or uncertainty tests.

5. P1 L30: What is the meaning of "...more sensitive to the aqueous reactions..." Please clarify.

**Response:** The daytime SOA mass was more influenced by aqueous phase reaction, compared to that in nighttime. To make it clear, the manuscript has been revised based on this comment.

L28: "The daytime SOA formation was generally more sensitive to the aqueous reactions than the nighttime SOA because the daytime chemistry produced more highly oxidized multifunctional products."

6. P1 L32: "Diurnal patterns" cover the whole daily cycle! Your experiments do not comply with this definition.

**Response:** The "Diurnal patterns" is replaced with the "sunlight intensity" in the revised manuscript according to this comment.

7. P2 L40: Organic aerosol is not a "well-known" factor – that is why research on this topic is so important. Please rephrase!

**Response:** "well-known" has been removed in the manuscript based on this comment.

8. P2 L42: SOA is in general important number wise, but for mass mostly in the PM1 range!

**Response:** To make it clear, the manuscript has been revised based on this comment.

L36: "A large portion of organic aerosol, especially of the fine particulate matter, is secondary organic aerosol (SOA) produced from the oxidation process of hydrocarbons (HCs), emitted from both biogenic and anthropogenic sources"

9. P2 L47: "The SOA from … is considerable in a global budget of SOA." Please rephrase, there is only one global SOA budget!

**Response:** The manuscript has been revised based on this comment.

L43: "Furthermore, the SOA from the oxidation of biogenic HCs is a considerable source of the global SOA budget (Kelly et al., 2018;Hodzic et al., 2016;Khan et al., 2017)."

10. P2 L50: "... HC is oxidized mainly with OH radicals ..." Is this still true in the presence of high concentrations of NOx?

**Response:** Yes. In daytime, the photolysis of $NO_3$ radical in the ambient air will be fast enough to consume the $NO_3$ radical with little reaction with hydrocarbons.

11. P2 L54: Ozone is normally not persistent during night times but destroyed by chemical reactions. Only when it is lifted above the boundary layer (e.g., by an inversion layer) can it survive until the next morning. Otherwise, tropospheric ozone concentration would continue to rise. Please rephrase!

**Response:** After sundown, $O_3$ is not rapidly and completely consumed as is the OH radicals. To make it clear, the manuscript has been revised based on the comment.

L51: "The $O_3$ generated in daytime is not rapidly consumed at nighttime and can react with $NO_2$ to form a $NO_3$ radical that can also be sustainable in nighttime."

12. P2 L64: What is "absorbing organic matter concentration"? Please clarify.

**Response:** This sentence has been moved to P2 L70 and relocated in the revised manuscript. Now reads,

L70: "The typical SOA models that are semi-empirically established a relationship between the organic matter (OM) concentration and the SOA yields by using simple partitioning parameters for two (Odum et al., 1996) or more surrogate products (Donahue et al., 2006) include organic-phase oligomerization, but they do not fully treat the SOA formation via the aqueous reactions in the presence of inorganic salts."

13. P2 L65: "The oxidation ... was approached by ...": Please rephrase.

**Response:** The manuscript has been revised based on this comment.

L59: "The biogenic SOA formation in current air quality models is predicted with the surrogate products originating from four major oxidants: OH radicals, $NO_3$ radicals, $O_3$, and $O(^3P)$."

14. P2 L67: What do you mean by "were not additive"?

**Response:** Due to the cross reactions, the gas phase reaction could be dynamic and hard to constrained by specific oxidation path. To make it clear, the manuscript has been revised based on this comment.

L60: "However, the gas phase reactions cannot be constrained by a specific oxidation path due to the various cross reactions with major oxidants."

15. P2 L74: "inorganic salted aqueous phase": An aqueous solution is by definition a solution of a chemical in water, often of an inorganic salt. I guess what you want to say is "aqueous phase of inorganic salts". This expression "salted aqueous phase or solution" is several times used throughout the manuscript. I recommend changing it.

**Response:** "inorganic salted aqueous phase" is replaced by "inorganic salt" or "aqueous phase of inorganic salts" in the revised manuscript.

16. P2 L78: "This model has been demonstrated by ..." I believe this sentence is incomplete.

**Response:** This sentence was moved to P2 L75 and revised based on this comment as below:

"This model was demonstrated by simulating the SOA formation from various aromatic HCs (Im et al., 2014;Zhou et al., 2019;Han and Jang, 2022), monoterpenes (Yu et al., 2021), and isoprene (Beardsley and Jang, 2016)."

17. P3 L 82: "...were generated from the four different ... pathways" How were they generated from the pathways? Rephrase!

**Response:** The model parameters were produced for the oxygenated products estimated by simulating the gas mechanisms under the various $NO_x$ levels (1~50 ppbC/ppb). To make it clear, the manuscript has been revised based on this comment as below:

"Lumping species and their stoichiometric coefficient and physicochemical parameters from the explicit gas mechanisms were individually obtained from the four different oxidation pathways with OH radicals, $O_3$, $NO_3$ radicals, and $O(^3P)$."

18. P3 L83: "To improve the feasibility ..., providing the HC consumption by each oxidant" This sentence is in parts an exact replication of the sentence in the introduction, and again it does not make too much sense. Please clarify. Also, a citation for the SAPRC07TC mechanism would appropriate.

**Response:** This sentence has been removed from the revised manuscript.

19. P3 L86: "The potential SOA yield(s) ... were applied" Applied to what? Please rephrase!

**Response:** The manuscript has been revised based on this comment.

L80: "The potential SOA yields of biogenic HCs via four different oxidation paths were simulated by using the UNIPAR model and utilized to study day and nighttime patterns in biogenic SOA formation under varying $NO_x$ levels, temperature, and seed conditions."

20. P3 L97: "... injected from a 2% NO cylinder under the air flow ..." What do you mean by under the airflow? Please clarify!

**Response:** The NO and $NO_2$ were injected from the cylinder through the clean air stream. To make it clear, the manuscript has been revised based on the comment.

L91: "NO was introduced into the chamber from the NO cylinder (2%, Air gas) prior to sunrise for daytime experiments. The $NO_x$ level is classified into the high $NO_x$ level (HC/$NO_x$ < 5.5 ppbC/ppb) and the low $NO_x$ level (HC/$NO_x$ > 5.5 ppbC/ppb) based on the initial concentrations of HC and NO."

21. P3 L99: How did you ascertain that the sulfuric acid seeds where not neutralized during injection? It is quite a task to prevent its neutralization by ammonia which is very abundant.

**Response:** Sulfuric acid could be neutralized during both injection and experiment. Thus, the inorganic concentrations ($SO_4^{2-}$, $NH_4^+$, and $NO_3^-$) of aerosol were monitored by using PILS-IC from the beginning of the experiment. The FS values, which is the $SO_4^{2-}$ to $NH_4^+$ ratio, were presented in the figures and Table 1, suggesting the aerosol acidity.

22. P3 L102: What do you mean with "NOx condition was controlled by $NO_2$"?

**Response:** To have the targeted $NO_x$ condition for the nighttime experiment, $NO_2$ was injected instead of the NO to produce $NO_3$ radical from the reaction between $NO_2$ and $O_3$. The sentence was rewritten to better understand for the reader.

L96: "$O_3$ was injected first into the chamber by using the $O_3$ generator (Jenesco Inc) followed by the $NO_2$ injection by using the $NO_2$ cylinder (2%, Air gas). Nighttime biogenic SOA formation was observed under the three different $NO_x$ levels (i.e., $O_3$ only, low $NO_x$ (HC/$NO_x$ > 5.5 ppbC/ppb) and high $NO_x$ level (HC/$NO_x$ < 5.5 ppbC/ppb))."

23. P3 L113: "The inorganic ion ... concentrations were in situ monitoring by ..." should be "... in situ monitored by a Particle-Into-Liquid-Sampler ..."

   **Response:** "monitoring " has been replaced with "monitored" in L107.

24. P3 L115: The use of butanol-based CPCs in chamber studies is quite controversial because of potential contamination. Did you check for this?

   **Response:** We aware this problem. After each chamber experiment, we disconnect SMPS from chamber. In addition, the UF-APHOR chamber is located on the roof of the Black Hall and thus, separated from the laboratory air. Prior to the chamber experiment, the chamber background air is monitored to ensure negligible contamination by other trace gases.

25. P3 L119: "An Aerosol Chemical Speciation Monitor ... to compare with data obtained from OC/EC and PILS-IC for accurate measurements." Each of these methods have their own uncertainties; therefore, I would avoid the term "accurate measurements".

   **Response:** "for accurate measurements" has been removed in this sentence.

26. P3 L119: "The relative humidity and temperature were monitored ..." How was this done (at how many different locations)? Since your chambers are quite large there could be a temperature (and humidity) gradient inside of it.

   **Response:** The relative humidity and temperature were monitored by the senser inside each chamber. However, the gradient of temperature or humidity inside of the chamber was not considered in this study.

27. P3 L120 (also P1 L9): "... sunlight intensity was measured by Total Ultraviolet Radiometer."? The results of these measurements are not presented in Table1 but could be important for the chemistry and the SOA mass yield!

   **Response:** The sunlight intensity can influence on the SOA mass yield as referee mentioned above. To indicate the sunlight intensity for each experiment, Table 1 has been updated in the revised manuscript as below:

| Exp. ID | Date | Initial condition | | | | | FS | Temp (K) | %RH | max OM ($\mu g\ m^{-3}$) | Max TUVR[4] ($W\ m^{-2}$) | Figures |
| | | HC (ppb) | HC/$NO_x$ (ppbC/ppb) | Seed[1] | Seed mass[2] ($\mu g\ m^{-3}$) | $OM_0$[3] ($\mu g\ m^{-3}$) | | | | | | |
|---|---|---|---|---|---|---|---|---|---|---|---|---|
| | | | | | Isoprene ($C_5H_8$) | | | | | | | |
| IS01 | 10/04/2021 | 750 | - | - | | 5 | | 295-302 | 44-81 | 27 | - | Fig. 2 (a) |
| IS02 | 10/07/2021 | 782 | 13.3 | - | | 5 | | 297-301 | 42-56 | 31 | - | Fig. 2 (b) |

| ID | Date | | | Seed[1] | Seed mass[2] | | FS | Temp. | RH | | | Figure |
|---|---|---|---|---|---|---|---|---|---|---|---|---|
| IS03 | 10/20/2021 | 750 | 3.9 | - | | 5 | | 292-298 | 36-75 | 30 | - | Fig. 2 (c) |
| IS04 | 12/16/2021 | 696 | 5.6 | - | | 5 | | 291-310 | 16-38 | 116 | 25.11 | Fig. 3 (a) |
| IS05 | 01/27/2015 | 839 | 17.4 | - | | 3 | | 279-298 | 27-66 | 62 | 25.81 | Fig. 3 (d) |
| α-pinene (C$_{10}$H$_{16}$) | | | | | | | | | | | | |
| AP01 | 03/19/2021 | 84 | - | - | | 4 | | 282-306 | 42-95 | 157 | - | Fig. 2 (d) |
| AP02 | 06/23/2021 | 92 | - | SA | 100 | 4 | 0.72 | 296-899 | 72-88 | 96 | - | Fig. 2 (g) |
| AP03 | 06/23/2021 | 79 | - | wAS | 100 | 4 | 0.33 | 296-300 | 89-100 | 80 | - | Fig. 2 (g) |
| AP04 | 09/09/2021 | 64 | 10.5 | - | | 5 | | 296-299 | 34-42 | 37 | - | Fig. 2 (e) |
| AP05 | 09/09/2021 | 58 | 10.5 | SA | 85 | 5 | 0.9-0.85 | 297-299 | 41-72 | 27 | - | Fig. 2 (e) |
| AP06 | 09/20/2021 | 61 | 3.7 | - | | 6 | | 297-301 | 37-55 | 28 | - | Fig. 2 (f) |
| AP07 | 09/20/2021 | 59 | 4.1 | SA | 87 | 6 | 0.85-0.75 | 298-302 | 37-55 | 33 | - | Fig. 2 (f) |
| AP08 | 11/04/2021 | 60 | 2.3 | dAS | 40 | 5 | 0.33 | 288-294 | 32-45 | 58 | - | Fig. 2 (h) |
| AP09 | 11/04/2021 | 60 | 2.3 | - | | 5 | | 289-293 | 44-66 | 63 | - | Fig. 2 (h) |
| AP10 | 08/28/2019 | 124 | 11.3 | - | | 4 | | 296-320 | 14-40 | 23 | 36.21 | Fig. 3 (b) |
| AP11 | 08/28/2019 | 130 | 10.7 | SA | 50 | 4 | 0.80-0.43 | 296-317 | 32-54 | 98 | 36.21 | Fig. 3 (b) |
| AP12 | 07/18/2019 | 142 | 4.9 | SA | 60 | 3 | 0.85-0.47 | 294-320 | 13-42 | 52 | 37.34 | Fig. 3 (e) |
| AP13 | 07/18/2019 | 139 | 4.6 | - | | 3 | | 294-319 | 19-48 | 28 | 37.34 | Fig. 3 (e) |
| β-caryophyllene (C$_{15}$H$_{24}$) | | | | | | | | | | | | |
| BC01 | 11/10/2021 | 50 | | | | 4 | | 292-299 | 29-67 | 95 | - | Fig. 2 (i) |
| BC02 | 11/10/2021 | 50 | | SA | 70 | 4 | 0.72 | 293-298 | 36-72 | 73 | - | Fig. 2 (i) |
| BC03 | 11/23/2021 | 40 | 4.2 | | | 3 | | 278-293 | 40-72 | 65 | - | Fig. 2 (j) |
| BC04 | 12/03/2021 | 50 | 10.5 | SA | 120 | 3 | 0.7-0.34 | 281-308 | 23-90 | 219 | 24.44 | Fig. 3 (c) |
| BC05 | 12/03/2021 | 50 | 10.5 | | | 4 | | 282-308 | 30-95 | 256 | 24.44 | Fig. 3 (c) |
| BC06 | 12/10/2021 | 50 | 3.8 | | | 3 | | 287-310 | 25-77 | 100 | 22.81 | Fig. 3 (f) |
| BC07 | 12/10/2021 | 50 | 3.8 | SA | 150 | 3 | 0.78-0.42 | 288-311 | 26-72 | 87 | 22.81 | Fig. 3 (f) |

[1]NS, SA, wAS, and dAS indicate non-seeded, sulfuric acid seed, wet ammonium sulfate seed, and dry ammonium sulfate seed, respectively. [2]The seed mass is determined as a dry mass, without water mass. [3]The pre-existing organic matter (OM$_0$) is determined for the chamber air prior to the injection of inorganic seed and HC. [4]Maximum sunlight intensity is shown during the experiment measured by using the TUVR. For nighttime, the experiment was performed under the darkness without the sunlight.

28. P3 L120: Why are the results of the aerosol acidity measurements not presented in Table 1? This could also help to answer the question regarding sulfuric acid neutralization.

**Response:** To present the aerosol acidity of each experiment, measured FS (fraction of sulfate = SO$_4^{2-}$/(SO$_4^{2-}$+NH$_4^+$)) ranges during the experiment has been added to the Table1 as shown above.

29. Follow up: In the beginning the authors speak of sulfuric acid seed particles, in the later parts they only mention ammonium bisulfate (which is to be expected). This should be clarified and correctly be mentioned right from the beginning.

**Response:** For the chamber study, sulfuric acid and ammonium sulfate have been used to demonstrate the neutral seed effect and the acidity effects on SOA formation. However, in the real

world, ammonium bisulfate could be the better inorganic seed to represent the ambient air. Thus, to implicate our model result to the real world, we simulate the ammonium bisulfate. To clarify this point, the manuscript has been revised by adding the sentence at P7 L243.

"To simulate the impact of aerosol acidity, the SOA formation is simulated in the presence of AHS seed, which is often found in ambient air. The reported acidity of the ambient aerosol is in the range of pH:-1~5 (Pye et al., 2020)."

30. P4 L124: Again, explicit to which degree? Which mechanisms were used (reference)?

**Response:** In this paragraph, the overall structure of the model was described. The detailed information of the explicit model and the further process of parameterization process is described in Sect. 3.1.

31. P4 L127: "… were determined by the near-explicit gas mechanism." Which mechanism?

**Response:** The sentence has been removed from the revised manuscript. The detailed information about the explicit mechanism has been given in the Sect. 3.1.

32. P4 L134: "… salted aqueous solutions …" How were they salted? Sodium chloride, ammonium sulfate, or something different?

**Response:** In this study, the salted aqueous solution includes only sulfate containing inorganics, such as sulfuric acid (SA), ammonium bisulfate (AHS), and ammonium sulfate (AS). The manuscript has been revised based on this comment.

L125: "For α-pinene and β-caryophyllene, the SOA formation in the presence of salted aqueous solutions (i.e., sulfuric acid (SA), ammonium bisulfate (AHS), and ammonium sulfate (AS)) was simulated under the assumption of the liquid-liquid phase separation (LLPS) between the organic and inorganic phase."

33. P4 L135: What is the relation between liquid-liquid-phase-separation and isoprene SOA formation and condensation on inorganic seed particles, and why did you exclude it? An explanation (and maybe a citation?) for the general reader would be nice.

**Response:** Isoprene SOA has high O:C ratio, indicating the high polarity, compared to other SOAs. Thus, several studies reported that the single homogenous mixed phase of the isoprene SOA in the presence of the inorganic seed. Thus, we conclude that there is a limitation to simulate isoprene SOA formation with our UNIPAR model due to the model assumption with LLPS. The manuscript has been revised to include this information in L128.

L128: "In case of isoprene, the production of single homogeneous mixed phase SOA has been reported in the presence of inorganic seed (Beardsley and Jang, 2016;Carlton et al., 2009). Thus, the isoprene SOA formation in the presence of inorganic aerosol was excluded."

34. P4 L147: What do you mean by "was included to fulfill the oxidation mechanism in the current regional model"?

**Response:** The oxidation process of biogenic HC by O($^3$P) is generally insignicifant path to produce SOA. However, in the regional scale model, biogenic HCs react with four major oxidants (OH radicals, O$_3$, NO$_3$ radicals, and O($^3$P)). To synchronize with the regional schale model, the oxidation path of biogenic HCs with O($^3$P) has been added. The manuscript has been revised.

L140: "Furthermore, the oxidation process of biogenic HCs by O(3P) (Paulson et al., 1992; Alvarado et al., 1998) was included to synchronize with the oxidation mechanism in the current regional model."

35. P4 L157: "ai ... was estimated by the predetermined mathematical equation originated from the explicit mechanism as a function of ..." This sentence is difficult to understand. What is "ai"? Are these the elements of the unified array calculated for each hydrocarbon oxidation pathway? Please elaborate!

**Response:** $\alpha_i$ is the stoichiometric coefficient of gas product (i) and it was defined in L124 and L133.

L124: "The HC consumption obtained from gas mechanisms, stoichiometric coefficient ($\alpha_i$), and physicochemical parameters of lumping species (i) were then applied to produce SOA mass (OM$_T$) via gas–particle partitioning (OM$_P$) and heterogeneous reactions (OM$_{AR}$) in both organic and inorganic phases."

L133: "The UNIPAR model utilizes the stoichiometric coefficient ($\alpha_i$) array and physicochemical parameters ($p_{L,i}^{\circ}$, $MW_i$, $O{:}C_i$, and $HB_i$) of the lumping species (i), which are determined by the explicitly predicted gas products."

36. P5 L169: OM$_T$ is only introduced on P6 L222.

**Response:** OM$_T$ is total organic matter, and it was defined in L124.

L124: "The HC consumption obtained from gas mechanisms, stoichiometric coefficient ($\alpha_i$), and physicochemical parameters of lumping species (*i*) were then applied to produce SOA mass (OM$_T$) via gas–particle partitioning (OM$_P$) and heterogeneous reactions (OM$_{AR}$) in both organic and inorganic phases."

37. P5 L 171: "... is also calculated as the traditional ..." should be "... according to the ..." or similar.

**Response:** The manuscript has been revised based on this comment.

"The partitioning coefficient of *i* into the inorganic phase ($K_{in,i}$, m$^3$ µg$^{-1}$) is also calculated according to the absorptive partitioning theory:"

38. P5 L209: Many phrases and statements are repeated again and again throughout the manuscript with nearly the exact wording, for instance here the "predetermined mathematical equations", and "lumping species generated from the explicit mechanism". The manuscript would gain a lot if those repetitions would be minimized, and the description of the mechanism development would be much more comprehensible for the reader.

**Response:** To reduce the repetitions in the manuscript, the sentence has been rephrased as below:

"RO$_2$ and HO$_2$ concentrations, and the consumptions of biogenic HCs are predicted with SAPRC07TC for four different oxidation pathways, and they are applied to predict the gas phase concentration of lumping species and SOA formation in UNIPAR."

39. P6 L230: "Thus, the seed effects observed in the presence of NO$_x$ ..." This sentence is really difficult to understand. Do you want to say that the presence of NOx had no influence on the SOA formation in the presence of seeds? What are those "seed effects"?

**Response:** Due to the hydrolysis of N$_2$O$_5$, the impact of inorganic seed on the nocturnal SOA formation was insignificant in the presence of NO$_x$. The manuscript has been revised based on this comment.

L223: "However, the impact of inorganic seed on nocturnal SOA formation can be insignificant in the presence of NO$_x$ because N$_2$O$_5$ undergoes heterogeneous hydrolysis reaction on the surface of wet aerosol particles to form nitric acid (HNO$_3$) (Brown et al., 2006;Hu and Abbatt, 1997;Galib and Limmer, 2021)."

40. P7 L243: Which large molecules and how did you measure them? And even if they have a poor solubility, couldn't the aerosol acidity still be important in a liquid organic phase?

**Response:** The physicochemical properties of gas products were not measured. The oxygenated products and their physicochemical properties were predicted from the explicit gas mechanism. The solubility of the gas products has been considered in the model with the O:C ratio, hydrogen bonding, and molecular weight based on the gas simulation results. Based on our simulation, the impact of acidic seed on β-caryophyllene SOA formation appeared but it was small due to their poor solubility. This sentence has been relocated in L296.

L296: "The large molecules originating from β-caryophyllene oxidation might have a poor solubility in aqueous phase, weakening the impact of aerosol acidity on OM$_{AR}$."

41. P7 L245: Why do you mention that isoprene products can be "mixed" with aqueous solutions of inorganic salts, when you did exclude this, as mentioned in the previous sentence?

**Response:** This sentence has been removed in the revised manuscript. Please find the response for the referee's comment #33.

42. P7 L270: First you mention that your results show that for isoprene OH is the most important oxidant and that this agrees with previous studies, then you write that reaction with ozone is only minor, only to conclude (again) that your results suggest that a sizable fraction (whatever that is) of isoprene SOM is formed via the OH pathway.

**Response:** Based on this comment, **"But the reaction with O$_3$ is a minor."** has been removed from the revised manuscript.

43. P7 L263: The whole paragraph about the α-pinene simulations has only one sentence that discusses your results. All remaining parts are statements and knowledge from references, which belong rather

into the introduction and not in a result/evaluation section if not put into a direct context with your simulations. What is missing here would be for example a discussion why for most cases there is no significant difference between low and high humidity (besides the ammonium bisulfate/acidic seed cases).

**Response:** The additional discussion has been added in the revised manuscript based on the comment.

"α-Pinene SOA yields are high with ozonolysis and $NO_3$-initiated oxidation in both daytime and nighttime. Low volatile products form from the autoxidation of ozonolysis products as reported in the previous studies (Roldin et al., 2019;Crounse et al., 2013;Bianchi et al., 2019). The addition of $NO_3$ to the alkene double bond of α-pinene is followed by the addition of an oxygen molecule to form an alkylperoxy radical that can also lead to low-volatile peroxide accretion products (ROOR) (Hasan et al., 2021;Bates et al., 2022). The α-pinene ozonlysis SOA yield is insensitive to humidity even in the presence of hygroscopic, acidic AHS seed. Unlike isoprene SOA (Beardsley and Jang, 2016) or aromatic SOA (Han and Jang, 2022;Im et al., 2014;Zhou et al., 2019), α-pinene gas products are relatively hydrophobic and thus, less soluble in the aqueous phase."

44. P8 L272: Have you investigated the amount of photo degradation of β-caryophyllene and the resulting products in your simulation to be able to calculate their volatility or is this a speculation or scientific knowledge (citation!)?

**Response:** The oxidation product from the β-caryophyllene is determined by simulating the explicit mechanisms and their physicochemical parameters were compared with that from other biogenic HCs. Based on the comment, the manuscript has been revised by adding the examples of the molecular weight of oxygenated products in reactive groups from three biogenic HCs.

L259: "In case of β-caryophyllene, even after the photodegradation the product volatility is still low enough to significantly partition to the aerosol phase and heterogeneously form SOA. Evidently, the molecular weight of β-caryophyllene oxidation products with high reactivity is generally higher than that in isoprene or α-pinene products. For example, the averaged molecular weight of β-caryophyllene oxidation products in highly reactive groups (VF or F) is 183.01 $g\ mol^{-1}$, while that from isoprene and α-pinene is 116.65 $g\ mol^{-1}$ and 143.29 $g\ mol^{-1}$, respectively."

45. P8 L279: Throughout the document you often equate NOx with the $NO_3$ radical, which can be done in parts at nighttime in the presence of ozone. However, during daytime $jNO_2$ might play an important role which could even terminate certain oxidation pathways (of VOCs) resulting in higher volatility organic nitrates in the gas phase and thus less SOA. Again, this paragraph consists in large parts of repetition and citation of literature without direct context to your results and does not really discuss the results of the simulation.

**Response:** In this section, SOA yields are simulated under the constrained oxidation path with a fixed amount of HC consumption to investigate the impact of product distributions of each oxidation path on SOA growth in day and night. Thus, this sentence describes the impact of NOx levels on the oxygenated product from each oxidation path in day and night. For the clear understanding of readers, the purpose of this section has been added in L237 of the revised manuscript.

L237: "The atmospheric process of biogenic HCs is complex because of their multi-generation oxidations by the combination of various oxidation paths. To investigate the impact of product

distributions of each oxidation path on SOA growth in day and night, SOA yields are simulated under the constrained oxidation path with a fixed amount of HC consumption as seen in Fig. 4."

46. P8 L303: "The simulation of ... is performed with ... In the presence of the chamber wall" – Do you mean to say that wall effects were included in the model? Please rephrase!

**Response:** Yes, the manuscript has been revised.

L293: "The simulation of SOA yields in Fig. 4 is performed with the model parameters obtained in the presence of the wall effects."

47. P8 L306: "Overall, the effect ..." Any suggestions why that could be the case?

**Response:** The impact of gas-wall deposition is more under the non-seeded conditions than that in the presence of the inorganic seeds. Thus, by correcting the wall loss, the increase of SOA mass is more in the absence of inorganic seed than that with inorganic seeds, resulting the less difference in SOA mass between the non-seeded and seeded conditions. The sentence has been added in L299 of the revised manuscript.

L299: "The impact of the chamber wall on SOA formation in the presence of inorganic seed was less than that without the wet inorganic seed (Krechmer et al., 2020;Han and Jang, 2022)."

48. P9 L317: Why did you choose those VOC concentrations? They seem quite high compared to typical values, especially in the urban atmosphere.

**Response:** For the chamber study, hydrocarbon and NOx concentrations are overall high due to the experimental limitations. Firstly, for the chamber study there are gas-wall loss and particle-wall loss which can influence on the experimental data. With those losses, high concentrations of hydrocarbons need to be injected to get enough mass which is above the detection limit or instrumental uncertainties. To have various $NO_x$ conditions, the target NOx concentrations were determined based on the HC/$NO_x$ ratio and needed HC concentration, resulting in the high concentration of $NO_x$ in the chamber. To indicate the influence of the initial concentration of biogenic HCs and NOx, the sensitivity of SOA model prediction to the initial HC concentration has been tested with three biogenic HCs and added to the SI.

49. P9 L318: "The sensitivity of SOA mass ... is simulated ..." should be "... is/was investigated by simulating ...". Why are the results of this simulation only shown in the supplementing material and not discussed here?

**Response:** The temperature sensitivity has been shown in the Fig. 5. Figures S2 illustrates the contribution of each oxidation path to the HC consumption to support Fig. 5. To avoid the confusion, those explanations moved to the early in the same paragraph (L308).

50. P9 L321: This paragraph repeats just the results presented in 4.2.

**Response:** The oxidation of biogenic HCs is associated with various oxidation paths. In Sect. 4.2, To investigate the impact of product distributions of each oxidation path on SOA growth in day and night,

SOA yields are simulated under the constrained oxidation path with a fixed amount of HC consumption. However, in Sect. 4.3, all four oxidation paths contribute to the biogenic SOA formation with different contributions for various environmental conditions. Thus, both section 4.2 and Sect. 4.3 have been modified to better understand for the reader.

51. P9 L324: "In addition, OH radical's contribution ... is positively correlated to ozonolysis, which produces OH radicals as a byproduct." This is a recursive statement and should be removed or rewritten.

**Response:** The sentence has been removed in the revised manuscript.

52. P9 L 342-345: This is nearly a 1:1 repetition of L316-319 ...

**Response:** This part is to describe the simulation condition used for the sensitivity test and thus, this part is needed. To reduce the repetition, the sentence has been updated in revised manuscript as below:

"The simulations are performed in the absence of the gas-wall partitioning (Han and Jang, 2022) with the same given initial concentration of isoprene, α-pinene, and β-caryophyllene with Fig. 5."

53. P9 L347: "In nighttime, the simulation suggests ... at the high NOx zone due to the high SOA potential ..." What is a NOx zone? Please rephrase sentence!

**Response:** "high NOx zone" has been replaced with "high NOx condition" in the revised manuscript as below:

54. P10 L360: What kind of gasoline fuel was used? And why this concentration? Is this relevant for the real atmosphere or is it just a simulation experiment. If it is relevant, what are common concentrations of gasoline fuel in the (urban?) atmosphere?

**Response:** US commercial gasoline fuel (octane number: 87) was used. Its composition has been reported previously (Han and Jang, 2022). Around 30% of gasoline fuel were aromatic compounds.

Both α-pinene and gasoline fuel were introduced to the chamber in the form of gas (Section 2). Both α-pinene and gasoline concentrations were higher than those in ambient air. As discussed in the response to question 24 from reviewer 1, the concentrations of chamber experiments are limited by detection of instruments. For our chamber studies, gasoline total carbon concentrations were nearly 3000 ppbC and those of α-pinene were about 800 ppbC (80 ppb). In the urban aera, the emissions are dominated by anthropogenic sources.

55. P10 L365: How was the oxidation of the gasoline fuel implemented? If it was not implemented comparable to for example the α-pinene (regarding mechanisms and chemistry) it could also explain why it did not contribute to the SOA formation in the simulation.

**Response:** The oxidation of aromatic hydrocarbons in gasoline fuel has been implemented by using the SAPRC07TC. The resulting HC (a-pinene and aromatic HCs) consumption, $RO_2$ concentration, and $HO_2$ concentration were applied to calculate the SOA formation with the SOA model parameters

from this study and Han and Jang, (2022). Thus, the SOA formation from the gasoline vapor has been counted in this study.

56. P10 L370: It has long been known that aromatic compounds can be efficient OH scavengers

**Response:** Yes, aromatic compounds have been known to be a OH scavengers. Aromatic compounds in the gasoline fuel also could be a SOA source. In this study, the goal of this section is to investigate the nocturnal SOA formation of α-pinene and the resulting SOA formation from the anthropogenic precursors.

57. P10 L381: The discussion of results from section 4.5 is effectively missing. It is hinted, that α-pinene SOA formation showed the strongest reaction to the changes of the model parameters, and that the change of the partitioning coefficient had the largest impact on the three biogenic SOA formation systems, but nothing more. Therefore, this section should be either extended or removed.

**Response:** This section is about the model uncertainties associated with the model parameters. This section has been relocated to the section 4.3 with the sensitivity test.

58. P10 L386: Why was a different relative humidity used in these simulations?

**Response:** The uncertainty test has been redone under the same relative humidity (RH =50%) with the sensitivity test.

59. P11 L394: Maybe "Conclusions" would be more fitting to this section!

**Response:** "Atmospheric Implications" is replaced with "Conclusions" in the revised manuscript.

60. P11 L411: "... under high NOx zones ..." should be "concentrations" or "levels".

**Response:** "high NOx zones: has been replaced with "high NOx levels" in the revised manuscript.

61. P11 L412: "Under the rural environment" should be "In rural environments" or "In a rural environment".

**Response:** "Under the rural environment" has been replaced with "In rural environments" in the revised manuscript.

62. P11 L415: Nearly all soluble inorganic and organic salts are electrolytes.

**Response:** Based on the comment, "Electrolytic inorganic salts" has been replaced with "Inorganic salts" in the revised manuscript.

63. P11 L419: "In this study ... showed a weak seed impact ..." How is this related to the previous statement about acidic seeds in other studies? Rephrase!

**Response:** The manuscript has been revised based on the comment as below:

"However, nighttime ozonolysis biogenic SOA in this study was insignificantly influenced by seed conditions as seen in the model simulation and chamber observations (Fig. 4)."

64. P11 L428: "... showing the higher biogenic HC emission" should be "showing higher biogenic HC emission(s)".

**Response:** The manuscript has been revised based on the comment as below:

"There is a diurnal pattern in the biogenic HC emission showing higher biogenic HC emissions during daytime (Holzke et al., 2006;Chen et al., 2020;Petron et al., 2001;Goldstein et al., 1998)."

65. P11 L429: Why is the "concentration of $NO_2$ generally high" during daytime? If there is no source of NO and no ozone, where is it supposed to come from? Rephrase the sentence!

**Response:** The manuscript has been revised based on the comment as below:

"The concentrations of $O_3$ and $NO_2$ are generally high in ambient air at daytime, involving the photochemical cycle of $NO_x$ in the presence of hydrocarbons."

66. P12 L435: "The model uncertainties ...mainly originate from gas mechanisms and aerosol phase reactions." What other factors are important for SOA formation besides gas and aerosol phase? Please rephrase.

**Response:** The manuscript has been revised based on the comment as below:

"The model uncertainties to predict SOA mass mainly originates from the simplified gas mechanisms and the missing aerosol phase reactions."

67. P12 L440: "Neither ... nor ... were not fully considered" In this context a double negation does not affirm but resolves to a positive. Rephrase!

**Response:** The manuscript has been revised based on the comment as below:

"Either complex cross reactions between $RO_2$ radicals or the long-term aging process of multiple generation products were not fully considered, which can be a source of the bias in SOA prediction."

68. P12 L447: The manuscript ends abruptly. There should be some final remarks or conclusions.

**Response:** The last paragraph of the manuscript has been revised to provide better flow and the conclusion and reads now,

"There are model uncertainties to predict SOA due to the simplified gas mechanisms and the missing aerosol phase reactions, although the UNIPAR model utilizes products originating from explicit gas mechanisms. For example, the daytime β-caryophyllene SOA of this study was underpredicted as seen in Fig. 3, suggesting that the improvement of explicit gas mechanisms is essential to better predict SOA formation. In the model, the multiphase reaction of biogenic HC is individually treated with four different oxidation paths. Either complex cross reactions between $RO_2$ radicals or the longterm aging process of multiple generation products were not fully considered, causing a bias in SOA prediction. In the presence of inorganic seed, heterogeneous hydrolysis of $N_2O_5$ was assumed to be very rapid. However, the variation of aerosol constituents can influence the accommodation coefficient of $N_2O_5$. For example, the heterogeneous hydrolysis of $N_2O_5$ on organic-coated aerosol can be slower than that in salted aqueous phase (Anttila et al., 2006). In addition, aerosol phase reactions such as hydrolysis of organonitrates and the oxidation of particulate OM were not included in the model. In the future, the performance of the UNIPAR model for the diurnal variation in biogenic SOA formation needs to be evaluated in regional scales (Yu et al., 2022)."

69. P17 Table 1: Why were such huge isoprene concentrations used for the experiments? I do not believe isoprene concentrations of 700 ppbC (140 ppbV) and above are of any atmospheric relevance. And even under such high hydrocarbon concentrations I would not consider 50 ppbV NOx to be a low NOx case!

**Response:**

For the chamber study, hydrocarbon and NOx concentrations are overall high due to the experimental limitations. Firstly, for the chamber study there are gas-wall loss and particle-wall loss which can influence on the experimental data. With those losses, high concentrations of hydrocarbons need to be injected to get enough mass which is above the detection limit or instrumental uncertainties. To have various $NO_x$ conditions, the target NOx concentrations were determined based on the HC/$NO_x$ ratio and needed HC concentration, resulting in the high concentration of $NO_x$ in the chamber. To indicate the influence of the initial concentration of biogenic HCs and NOx, the sensitivity of SOA model prediction to the initial HC concentration has been tested with three biogenic HCs and added to the Sec. S5.

70. P17 Table 1: I am quite surprised by mass yields as high as 157 µg per cubic meter from 84 ppbC (8.4 ppbV) a-pinene, especially without any seed (AP01).

**Response:** The unit for the hydrocarbon concentration was wrong in the Table 1. Table 1 has been revised. Please find the revised table in the response of the comment# 27.

71. P20 Figure 2: The caption claims to present $OM_T$ and $OM_{AR}$ (dotted line), while the legend annotates the dotted line as $OM_P$

**Response:** The caption has been revised based on the comment. the dotted line is $OM_P$.

72. P22 Figure 4: It is difficult to distinguish between $O(^3P)$ and OH

**Response:** The SOA formation from the oxidation of biogenic HCs with $O(^3P)$ is insignificant in this simulation. In Fig. 4, SOA yield from $O(^3P)$ oxidation path is not visible.

73. P23 Figure 5: All plots are too small. In addition, the axis of (a) suppresses the view of the dynamics in the plot. While it is in general a good idea to scale plots similar this should not hinder the interpretation of the diagrams.

**Response:** Figure 5 has been revised with large size of the graph and the scale of the Fig. 5(a) has been updated based on this comment.

74. P23 Caption: What does "the reference sunlight intensity" mean? Is this the total intensity of the sunlight at that specific day, or is it a wavelength dependent intensity (plot), and why did you choose this specific date? How does it compare to the average sunlight intensity at this place, and at other places? This is a very arbitrary measure which modern science tries to and should avoid. Please reason why you chose this day and intensity spectrum.

**Response:** The reference sunlight intensity has been used for the sensitivity (Figs. 5 and 6) and uncertainty (Fig. S5) tests. The reference sunlight intensity measured on 06/19/2015 near summer solstice in the UF-APHOR illustrated in Fig. S1.

75. P24 Figure 6: Like Figure 5 all diagrams are way too small. Again, the scaling of the y-axis of the isoprene plot makes it impossible to clearly see any dynamic factors.

**Response:** Figure 6 has been updated based on the comment in the revised manuscript.

**References**

Anttila, T., Kiendler-Scharr, A., Tillmann, R., and Mentel, T. F.: On the reactive uptake of gaseous compounds by organic-coated aqueous aerosols: Theoretical analysis and application to the heterogeneous hydrolysis of N2O5, The Journal of Physical Chemistry A, 110, 10435-10443, 2006.
Han, S., and Jang, M.: Prediction of secondary organic aerosol from the multiphase reaction of gasoline vapor by using volatility–reactivity base lumping, Atmospheric Chemistry and Physics, 22, 625-639, 2022.
Krechmer, J. E., Day, D. A., and Jimenez, J. L.: Always Lost but Never Forgotten: Gas-Phase Wall Losses Are Important in All Teflon Environmental Chambers, Environmental Science & Technology, 54, 12890-12897, 2020.

---

## Author Comment (AC2)

**Response to the Referee2 (Manuscript Ref. NO.: acp-2022-327)**

We would like to thank the reviewer for their time, and useful comments. Their comments are repeated below, followed by our response.

**Comment on "Modeling Diurnal Variation of SOA Formation via Multiphase Reactions of Biogenic Hydrocarbons" Anonymous Referee #2**

**General Comments**

Han et al present a series of experiments conducted in a rooftop chamber examining the oxidation of three biogenic hydrocarbons (isoprene, a-pinene, b-caryophyllene) during both daytime and nighttime conditions. They examine the role of four different oxidants (OH,  $O_3$ ,  $NO_3$ ,  $O(^{3}P)$ ) and a series of environmental conditions, including hydrocarbon to NOx levels, relative humidity, temperature, and particle seed composition.

The major emphasis of the paper is on a gas-particle partitioning model, UNIPAR, that is first fit to the experimental data and then used to make predictions about the variation in SOA yields with different parameters. Major conclusions are that there is a strong positive  $NO_x$  dependence to SOA yield during nighttime conditions and a weaker negative  $NO_x$  dependence during daytime, and that there is a modest negative temperature dependence.

Overall, the paper is in line with other studies of these systems in the recent literature, but offers some new insights based on the explicit gas-particle partitioning model. Some aspects of the presentation should be clarified prior to publication, however, as outlined in the more specific comments below.

**Specific Comments:**

1. Line 43: Give the total SOA budget for reference. Also add the caveat in this line that these are models of global SOA, and that the cited work is just one of several estimates of this quantity.

**Response:** To clarify this point, global SOA production rates and additional citation were added in the revised manuscript.

L43: "Furthermore, the SOA from the oxidation of biogenic HCs is a considerable source of a global budget of SOA (Kelly et al., 2018; Hodzic et al., 2016;Khan et al., 2017). For example, Kelly et al. (2018) reported that the more than 50% of the annual global SOA production rates (48.5-74.0 Tg SOA yr-1) is from monoterpenes (19.9 Tg SOA yr-1) and isoprene (4-19.6 Tg SOA yr-1)."

2. Line 54: Not clear what is meant by a sustainable NO3 radical – perhaps this refers to production of NO3 radicals being sustained?

**Response:** This sentence refers to the production of  $NO_3$  radicals being sustained as referee commented. The manuscript has been revised.

L51: "The  $O_3$  generated in daytime is not rapidly consumed at nighttime and can react with  $NO_2$  to form a  $NO_3$  radical that can also be sustain in nighttime."

3. Line 58: There are more recent references to the organic nitrate yield from NO3 + isoprene. See for example:

Brownwood, B., A. Turdziladze, T. Hohaus, R. Wu, T.F. Mentel, P.T.M. Carlsson, E. Tsiligiannis, M. Hallquist, S. Andres, L. Hantschke, D. Reimer, F. Rohrer, R. Tillmann, B. Winter, J. Liebmann, S.S. Brown, A. Kiendler-Scharr, A. Novelli, H. Fuchs, and J.L. Fry, Gas-Particle Partitioning and SOA Yields of Organonitrate Products from NO3-Initiated Oxidation of Isoprene under Varied Chemical Regimes. ACS Earth and Space Chemistry, 2021. 5(4): p. 785-800.

Perring, A.E., A. Wisthaler, M. Graus, P.J. Wooldridge, A.L. Lockwood, L.H. Mielke, P.B. Shepson, A. Hansel, and R.C. Cohen, A product study of the isoprene+NO3 reaction. Atmos. Chem. Phys., 2009. 9(1): p. 4945-4946.

Response: Those works were added to the revised manuscript.

"For example, the oxidation of isoprene with the NO3 radical can rapidly produce nitrate containing products, resulting the increase in the SOA formation, up to 80% of gas products from the isoprene-NO3 oxidation (Kwok et al., 1996; Barnes et al., 1990;Perring et al., 2009;Brownwood et al., 2021)."

4. Line 97: The definitions of high and low NOx seem arbitrary and as though they might both be high NOx. Was the fate of  $RO_2$  radicals considered in defining the high and low NOx conditions – i.e., the rate of  $RO_2$  + NO compared to other  $RO_2$  losses?

**Response:** The high NOx and low NOx condition has been defined based on the HC/NOx ratio as HC/NOx < 5 ppbC/ppb and HC/NOx > 5 ppbC/ppb, respectively (in Sect. 2). The fate of RO2 radicals is covered by the explicit gas mechanism and the resulting stoichiometric coefficient arrays, which are function of HC/NOx ratios and aging (Sect. 3.1).

5. Line 145: Inclusion of O(3P) is relatively unusual and not normally important in the lower atmosphere (also a conclusion of this study). What motivated the inclusion of this oxidant rather than other minor oxidants such as chlorine radicals or Criegee intermediates?

**Response:** To increase the simplicity and the applicability of UNIPAR model in regional scale, SAPRC07TC has been integrated with the UNIPAR model. In the SAPRC07TC, the oxidation processes of biogenic HCs were treated with 4 different oxidants (i.e.,  $O_3$ , OH radicals,  $O(^{3}P)$ , and  $NO_3$  radicals). Thus, four different oxidation paths with those oxidants were considered in this study. To make it clear, a sentence below has been added in L140.

L140: "Furthermore, the oxidation process of biogenic HCs by  $O(^{3}P)$  (Paulson et al., 1992;Alvarado et al., 1998) was included to synchronize with the oxidation path in the current regional model."

6. Line 214: At what rate was N2O5 hydrolysis included, and how efficiently does this compete with gas phase NO3 reactions?

**Response:** The hydrolysis rate constant of  $N_2O_5$  was set as  $10^{-2}$  s-1, by simulating the chamber generated SOA data. The hydrolysis rate constant of  $N_2O_5$  has been reported as in a range of  $10^{-7} \sim 10^{0}$  s-1 (Wagner et al., 2013;Wood et al., 2005). The sentence below has been added to the revised manuscript.

L207: "The hydrolysis rate constant of  $N_2O_5$  has been reported as in a range of  $10^{-7} \sim 10^0$  s-1 (Wagner et al., 2013; Wood et al., 2005) and thus, the hydrolysis rate constant of  $N_2O_5$  was set to  $10^{-2}$  s-1 in this study."

7. Figure 2: The abbreviations NS, SA, wAS, etc. are not defined in the figure or the caption and not easy to find in the text. Clarify the meaning of these abbreviations in the figure.

**Response:** To make it clear, the definitions of seed condition in the chamber has been added in the figure captions.

"NS, SA, wAS, and dAS indicate non-seeded, sulfuric acid seeded, wet ammonium sulfate seeded, and dry ammonium sulfate seeded experiment, respectively."

8. Line 230: Conclusion not clear in this sentence. Is this stating that in the presence of aerosol there is no NO3 reaction with the biogenic hydrocarbons?

**Response:** There was an insignificant seed effect in nocturnal SOA formation in the presence of  $NO_x$  due to the removal of  $NO_3$  radical via heterogenous hydrolysis of the  $N_2O_5$ . However, a small increase in SOA mass was shown in several experiments and it could be a result of the aqueous phase reaction of the ozonolysis products. To clarify this point, the manuscript has been revised as below:

L222: "However, the impact of inorganic seed on nocturnal SOA formation can be insignificant in the presence of  $NO_x$  because  $N_2O_5$  undergoes heterogeneous hydrolysis reaction on the surface of wet aerosol particles to form nitric acid (HNO3) (Brown et al., 2006;Hu and Abbatt, 1997;Galib and Limmer, 2021). The small increase in SOA formation by inorganic seed is mainly caused by the aqueous phase reaction of the ozonolysis products."

9. Line 248: The biogenic mixing ratios used in the simulations are unrealistically large does this also bias the SOA yields high?

**Response:** Possibly, yes. To see the sensitivity of the SOA model prediction associated with the environmental parameters, consumption of biogenic HC is needed to be high enough. Based on the referee's comment, the sensitivity of the biogenic SOA formation to the initial HC concentration has been tested and added to the SI section 5. A paragraph has been added in L352.

"For the chamber study, concentrations of HC and NOx are generally higher than those in ambient air due to the detection limit of analytical instruments. Additionally, the chamber-generated SOA data can be influenced by vapor-wall deposition and the particle-wall loss. Fig. S4 illustrates the influence of the initial concentration of biogenic HCs on SOA formation at a given NOx level (high NOx condition). Regardless of initial HC concentrations, the sensitivity of SOA yields to different light conditions (day vs. night) or seed conditions (non-seed vs. wAHS) is consistent at a given biogenic hydrocarbon."

10. Line 254: SOA yields from NO3 said to be low during daytime, but Figure 4 shows them to be larger than OH? Is this correct? The description of isoprene SOA beginning in this line does not appear consistent with what appears in the figure.

**Response:** Figure 4 illustrates the potential SOA formation when the same amount of HC is consumed by each oxidation path. In the daytime, SOA yield is still high if the isoprene is consumed by  $NO_3$  radical as seen in Fig. 4. However, the rapid photolysis of  $NO_3$  radical can reduce the contribution of oxidation path through the biogenic HC +  $NO_3$  radical. To clarify this point, the manuscript has been revised as below:

L237: "The atmospheric process of biogenic HCs is complex because of their multi-generation oxidations by the combination of various oxidation paths. To investigate the impact of product distributions of each oxidation path on SOA growth in day and night, SOA yields are simulated under the constrained oxidation path with a fixed amount of HC consumption as seen in Fig. 4."

L245: "For isoprene, the efficient pathways to form SOA are the NO3-initiated oxidation (6-17%) and OH-initiated oxidation (3 - 4%) in both day and night at given conditions of Fig. 4."

11. Line 279: This paragraph contains a series of qualitative statements about the roles of different mechanistic pathways in forming SOA. Presumably, all of these could be quantified with the model and shown as a figure?

**Response:** The UNIPAR model can quantify SOA formation at a given mechanism under the controlled environmental condition. Figure 4 was simulated SOA yields under the constrained oxidation path with a fixed amount of HC consumption. In order to provide better understanding of the manuscript to the reader, the section 4.2 was modified in the revised manuscript. The 1st paragraph of section 4.2 reads now,

"The atmospheric process of biogenic HCs is complex because of their multi-generation oxidations by the combination of various oxidation paths. To investigate the impact of product distributions of each oxidation path on SOA growth in day and night, SOA yields are simulated under the constrained oxidation path with a fixed amount of HC consumption as seen in Fig. 4. SOA yields simulated under varying environmental conditions including two different NOx levels ((a) high NOx: HC/NOx = 3 ppbC/ppb and (b) low NOx: HC/NOx = 10 ppbC/ppb) and three different seed conditions (no seed, wAS, and wet ammonium bisulfate (wAHS)). To simulate the impact of aerosol acidity, the SOA formation is simulated in the presence of AHS seed, which is often found in ambient air. The reported acidity of the ambient aerosol is in the range of pH:-1~5 (Pye et al., 2020). Overall, biogenic SOA formation from the O(3P) reaction path is negligible."

12. Line 362: What is the chemical composition of gasoline fuel? Presumably this is in the gas phase? Is the high mixing ratio used here realistic to ambient conditions?

**Response:** US commercial gasoline fuel (octane number: 87) was used. Its composition has been reported previously (Han and Jang, 2022). Around 30% of gasoline fuel were aromatic compounds.

Both  $\alpha$ -pinene and gasoline fuel were introduced to the chamber in the form of gas (Section 2). Both  $\alpha$ -pinene and gasoline concentrations were higher than those in ambient air. As discussed in the response to question 24 from reviewer 1, the concentrations of chamber experiments are limited by detection of instruments. For our chamber studies, gasoline total carbon concentrations were nearly 3000 ppbC and those of  $\alpha$ -pinene were about 800 ppbC (80 ppb). In the urban aera, the emissions are dominated by anthropogenic sources.

13. Line 403: Suggest removing the reference to "government agency" and preferring instead to NOx control measures.

**Response:** The manuscript has been revised based on this comment as below:

"The  $NO_x$  emission from anthropogenic sources has gradually decreased, and it impacts the  $NO_3$  concentrations."

14. Line 415: The term "electrolytic" appears out of place here.

**Response:** "electrolytic" has been removed from the revised manuscript.

**References**

Han, S., and Jang, M.: Prediction of secondary organic aerosol from the multiphase reaction of gasoline vapor by using volatility–reactivity base lumping, Atmospheric Chemistry and Physics, 22, 625-639, 2022.

Wagner, N., Riedel, T., Young, C. J., Bahreini, R., Brock, C. A., Dubé, W., Kim, S., Middlebrook, A., Öztürk, F., and Roberts, J.: N2O5 uptake coefficients and nocturnal NO2 removal rates determined from ambient wintertime measurements, Journal of Geophysical Research: Atmospheres, 118, 9331-9350, 2013.

Wood, E., Bertram, T., Wooldridge, P., and Cohen, R.: Measurements of N 2 O 5, NO 2, and O 3 east of the San Francisco Bay, Atmospheric Chemistry and Physics, 5, 483-491, 2005.

---

## Referee Report (RR1)

**Review of "Modeling Day and Nighttime SOA Formation via Multiphase Reactions of Biogenic Hydrocarbons"**

Overall Comment:

This manuscript uses both chamber experiments and modeling to explore photochemical and nocturnal production of SOA from three different biogenic VOC precursors.  The approach and questions asked are scientifically relevant to the readership of ACP.  However, I have several concerns about the manuscript's readability, overstatement of some conclusions, and missing methodological details that should be addressed before publication.

General remarks:

Understanding the mechanistic processes that contribute to SOA yields is an important question for the atmospheric chemistry community.  The authors use a nice general approach of using chamber experiments to validate their model and then using their model to explore mechanistic details of the chemistry.

However, the results/discussion section is very hard to follow.  As Referee 1 noted in their remarks, "some parts of the manuscript seem to be more a review paper than an original work." I think this sentiment still applies to the results/discussion section.  The original conclusions of this work are deeply buried and hard to follow in this section.  I recommend  revision of this section to make the conclusions more easily identifiable to the reader.  Moreover, many of the mechanistic conclusions stated in the paper are backed up only by previous literature, rather than a detailed analysis of the authors' modeling work.  Including some process-level modeling results to support the stated conclusions would greatly improve the science presented (some specific examples of this are listed below as "specific comments").

Similarly, the abstract is very long-winded and makes it hard to understand the science question that the authors are trying to answer.  I recommend revising the abstract to focus more on the big picture impact of this work rather than a list of detailed conclusions.

Some methodological details are missing, including: Was the chamber filled with outdoor air or zero air prior to perturbation by VOC, seed, NOx, and O3 injections?  If filled with outdoor air, how did the composition vary between experiments?  How was the chamber cleaned between experiments?

Figure S2 seems like an important figure that would be worth considering moving to the main text since it relates to some of the main conclusions about the work involving the interaction of different oxidants.

Because the chamber used in this study is outside, the experiments presented here are not explicitly 'controlled experiments,' i.e., multiple variables change between each experiment. This variability is fine when using these experiments to validate their model since the variability is captured in the model inputs, and a framing that emphasizes that the model can capture the observed variability is useful.  However, it does become problematic when drawing process-level

conclusions in Section 4.1 since variation in multiple variables could be contributing to the observed trends. As such, I think some of the conclusions in Section 4.1 are overstated.

Lastly, the paper contains numerous grammatical errors that inhibit its readability. I recommend thorough proofreading.

Specific comments:

Lines 15-19: These three sentences are hard to understand and should be rewritten for clarity.

Lines 39-41: This sentence contains redundant statements about the dominance of biogenic HC emissions.

Lines 51-52: Note that $O_3$ titration is possible in some environments.

Line 54: It's unclear what the quoted 80% refers to.

Section 2: Referencing Table 1 in this section would be helpful to the reader.

Line 101: What was the $CCl_4$ from?

Line 124: What process does the "stoichiometric coefficient" refer to? Is it for each lumped bin?

Line 141-142: What model does "the current regional model" refer to?

Line 148: It could be helpful to include the $C*$ saturation vapor pressure equivalents for the vapor pressures listed here.

Line 149: Can you give quantitative descriptions for the reactivity scale?

Section 3.4 and Figure 1: I'm confused by which parts of the model mechanism uses MCM gas-phase chemistry and which parts use SAPRC chemistry. Please clarify.

Fig. 2(a): Why isn't there a line for $OM_p$ here?

Fig. 2: An indication of when t=0 is (e.g., injection time at night or sunrise in the daytime) would be helpful in these plots. Additionally, I recommend noting in the caption that meteorological parameters (e.g., temperature, relative humidity) varied between these cases in addition to the chemical parameters already noted. It might also be more appropriate to label the middle and right columns as "VOC + $O_3$ + $NO_3$" instead of just "VOC + $O_3$".

Line 225: This conclusion about an increase in SOA formation because of aqueous reactions of ozonolysis products is unsupported. Can you include a figure that shows process-level detailed analysis of the model to support this?

Line 242-243: Reporting the atmospheric range of pHs would be more useful if the manuscript also included information about the pH of the seeds used in the experiments.

Line 248-250: The sentence beginning with "Evidently…" is redundant.

Lines 258-259:  The conclusion about the impact of photolysis of isoprene and a-pinene oxidation products on SOA production is unsupported.

Lines 261-263: Why is the high reactivity relevant here?

Lines 283-285: The speculation presented here can be supported by the modeling work in this paper.  Including supporting plots/calculations would be useful.

Line 316: Is the difference between nighttime and daytime SOA for the same amount of initial VOC?  Does this statement account for variation in emissions / PBLH between day and night?

Line 320: Note that $NO_3$ can also be lost at night by reaction with fresh NO emissions.

Lines 371-373: How does the ratio of alpha-pinene to gasoline correspond to that in the atmosphere?

Lines 383-385:  Can you provide a plot to support the conclusion of this sentence, or at least a quantification of the conclusion presented here?

Fig 3(f): Why is there no dotted black line in Figure 3(f)?

Fig 4:  Because RH=45% is right around the efflorescence point for ammonium sulfate, do these high and low RH cases actually probe significant differences in aerosol liquid water content? Also, for clarity, I recommend noting in the caption that this figure presents modeling results.

Fig 7: What is driving the oscillations in the fraction of SOA from alpha pinene (solid red line)?

---

## Editor Decision (ED1)

Thank you very much for addressing most of the reviewer's comments, the manuscript is greatly improved. After minor revisions regarding previous referee comments, I am happy to accept the manuscript for publication.

**Referee #3** - Similarly, the abstract is very long-winded and makes it hard to understand the science question that the authors are trying to answer. I recommend revising the abstract to focus more on the big picture impact of this work rather than a list of detailed conclusions.

**Response:** The abstract has been revised based on the referee's comment.

**Review by Editor**: The track-changes file shows hardly any changes to the abstract, please address the reviewer's comment.

Referee #3 - Line 149: Can you give quantitative descriptions for the reactivity scale?

**Response**: We reported the mathematical description of model parameters including reactivity scales and physicochemical parameters, and mathematical equations for stoichiometric coefficients in the recently published paper by Yu et al (Supporting Information of ACP, 2022). In this manuscript, same reactivity scale has been used, and the physicochemical parameters and mathematical equations for stoichiometric coefficients are extended to biogenic HCs at four major oxidation paths to simulate day and night SOA mass as seen in Sect. S7.

**Review by Editor**: Please provide a reference to Yu et al. around line 149.

**Referee #3** - Section 3.4 and Figure 1: I'm confused by which parts of the model mechanism uses MCM gas phase chemistry and which parts use SAPRC chemistry. Please clarify.

**Response**: Please find the response to comment 7. The model parameters and the predetermined mathematical equations for stoichiometric coefficients for lumping bins were derived by using the product predicted from the semi-explicit mechanisms for the atmospheric oxidation of biogenic HCs. The model parameter and the equations are integrated to the predicted hydrocarbon consumption from any gas mechanisms. In order to support SOA formation in complex ambient air, model parameters and equations for stoichiometric coefficients were integrated with SAPRAC07TC. The hydrocarbon consumption predicted with both semi-explicit mechanisms and SAPRAC07TC well accords with that observed in chamber studies.

**Review by Editor**: I would like to reiterate on referee #3's question. When was MCM used and when was SAPRC used? Please refer to any previously published work here when determination of mathematical equations for stoichiometric coefficients for lumping bins was performed for a previous study and please outline the exact methodology (incl. showing calculation results) when it was newly performed for this study.

---

## Author Response (AR2)

**Modeling Day and Nighttime SOA Formation via Multiphase Reactions of Biogenic Hydrocarbons (Manuscript Ref#: acp-2022-327)**

**Response to the Referees and editor**

We would like to thank the referees and editor for their time and thoughtful comments on this manuscript. Their comments are repeated below followed by our response to each comment.

**Comment from Anonymous referee #1**

The revised manuscript has improved a lot from the previous version, and merits publication in ACP.

However, before publication the authors should still address a few issues.

1. I recommend replacing the term "explicit mechanism" with "extended semi-explicit mechanism" and to detail the extension/changes/additions made to the mechanism in the manuscript (e.g., "using the semi-explicit Master-Chemical-Mechanisms extended by ... "). A real "explicit mechanism" would include all possible reactions and pathways (including minor channels) and thus result in millions (if not billions) of calculations.
   **Response:** The word "explicit mechanism" has been replaced with "extended semi-explicit mechanism" in the revised manuscript.

2. Minor comment: in L240, rev. manuscript: "SOA yields simulated ..": This sentence is missing a verb. Maybe a final proof reading of the manuscript might be helpful.
   **Response:** This sentence has been modified in the revised manuscript.
   "SOA yields are simulated under varying environmental conditions including two different $NO_x$ levels ((a) high $NO_x$: HC/$NO_x$ = 3 ppbC/ppb and (b) low $NO_x$: HC/$NO_x$ = 10 ppbC/ppb) and three different seed conditions (no seed, wAS, and wet ammonium bisulfate (wAHS))."

**Comment from Anonymous referee #3**

**Overall Comment:**

This manuscript uses both chamber experiments and modeling to explore photochemical and nocturnal production of SOA from three different biogenic VOC precursors. The approach and questions asked are scientifically relevant to the readership of ACP. However, I have several concerns about the manuscript's readability, overstatement of some conclusions, and missing methodological details that should be addressed before publication.

**General remarks:**

Understanding the mechanistic processes that contribute to SOA yields is an important question for the atmospheric chemistry community. The authors use a nice general approach of using chamber experiments to validate their model and then using their model to explore mechanistic details of the chemistry.

However, the results/discussion section is very hard to follow. As Referee 1 noted in their remarks, "some parts of the manuscript seem to be more a review paper than an original work." I think this sentiment still applies to the results/discussion section. The original conclusions of this work are deeply buried and hard to follow in this section. I recommend revision of this section to make the conclusions more easily identifiable to the reader. Moreover, many of the mechanistic conclusions stated in the paper are backed up only by previous literature, rather than a detailed analysis of the authors' modeling work. Including some process-level modeling results to support the stated conclusions would greatly improve the science presented (some specific examples of this are listed below as "specific comments").

**Response:** Please find the responses for the specific comments below.

Similarly, the abstract is very long-winded and makes it hard to understand the science question that the authors are trying to answer. I recommend revising the abstract to focus more on the big picture impact of this work rather than a list of detailed conclusions.

**Response:** The abstract has been revised based on the referee's comment.

Some methodological details are missing, including: Was the chamber filled with outdoor air or zero air prior to perturbation by VOC, seed, $NO_x$, and $O_3$ injections? If filled with outdoor air, how did the composition vary between experiments? How was the chamber cleaned between experiments?

**Response:** The chamber was cleaned for 2 days after ventilation with outdoor air. Thus, the background is near zero air after cleaning. This point has been added to the revised manuscript.

L86: "Before experiment, chamber was cleaned by using clean air generator for 2 days after ventilation process."

Figure S2 seems like an important figure that would be worth considering moving to the main text since it relates to some of the main conclusions about the work involving the interaction of different oxidants.

**Response:** Figure S2 has been moved to the main body based on the referee's comment.

Because the chamber used in this study is outside, the experiments presented here are not explicitly 'controlled experiments,' i.e., multiple variables change between each experiment. This variability is fine when using these experiments to validate their model since the variability is captured in the model inputs, and a framing that emphasizes that the model can capture the observed variability is useful. However, it does become problematic when drawing process-level conclusions in Section 4.1 since variation in multiple

variables could be contributing to the observed trends. As such, I think some of the conclusions in Section 4.1 are overstated.

**Response:** Based on the referee's comment, some conclusions were moved to the sections 4.2.

Lastly, the paper contains numerous grammatical errors that inhibit its readability. I recommend thorough proofreading.

**Response:** The manuscript has been revised.

**Specific comments:**

1. Lines 15-19: These three sentences are hard to understand and should be rewritten for clarity.
   **Response:** These three sentences are modified in the revised manuscript.
   L13: "The resulting oxygenated products were then classified into volatility-reactivity based lumping species. The stoichiometric coefficients associated with lumping species were dynamically constructed under varying $NO_x$ levels and aging scales and they were applied to the UNIPAR SOA model."

2. Lines 39-41: This sentence contains redundant statements about the dominance of biogenic HC emissions.
   **Response:** This sentence has been removed.

3. Lines 51-52: Note that $O_3$ titration is possible in some environments.
   **Response:** A sentence has been added to the revised manuscript.
   L48: "In the presence of $O_3$, NO is titrated to form $NO_2$."

4. Line 54: It's unclear what the quoted 80% refers to.
   **Response:** To make it clear, the sentence has been modified in the revised manuscript.
   L50: "For example, the oxidation of isoprene with the NO3 radical can rapidly produce nitrate-containing products (up to 80% of total gas products) resulting the increase in the SOA formation (Kwok et al., 1996;Barnes et al., 1990;Perring et al., 2009;Brownwood et al., 2021)."

5. Section 2: Referencing Table 1 in this section would be helpful to the reader.

   **Response:** A sentence below has been added to the end of this section to refer the Table 1.
   L98: "The detail information of the chamber experiments is summarized in Table 1."

6. Line 101: What was the $CCl_4$ from?
   **Response:** $CCl_4$ was injected into the chamber with HCs to measure the chamber dilution.
   L 97: "$CCl_4$ was also introduced to the chamber to measure chamber dilution."

7. Line 124: What process does the "stoichiometric coefficient" refer to? Is it for each lumped bin?
   **Response:** Yes, stoichiometric coefficients are for each lumped bin. These stoichiometric coefficients were obtained by using oxidation products predicted from the semi-explicit mechanism and they are applied to the SOA model.
   L121: "The stoichiometric coefficient ($\alpha_i$) and physicochemical parameters of lumping species ($i$) are estimated by using the products predicted from extended semi-explicit mechanisms at a given oxidation path for each precursor. In the model, $\alpha_i$ values are dynamically constructed with a mathematical

equation as a function of NO$_x$ levels and gas aging. The predetermined mathematical equation and physicochemical parameters for lumping groups are applicable to the conventional gas mechanisms. In order to support the atmospheric oxidation of biogenic HCs in complex ambient air, these model parameters were integrated with the HC consumption predicted by the Statewide Air Pollution Research Center (SAPRC07TC) (Carter, 2010) gas mechanisms and then applied to produce SOA mass"

8.  Line 141-142: What model does "the current regional model" refer to?
    **Response:** Here, the regional scale models refer to the regional scale models, such as CMAQ, CAMx, Geos-Chem, and WRF-Chem, which utilize conventional ozone model (Carbon Bond (CB) mechanism or SAPRC) to simulate the HC consumption. Those CB or SAPRC mechanisms treat the oxidation process of biogenic HCs with major atmospheric oxidants. To make it clear, the sentence has been modified in the revised manuscript
    L144: "Furthermore, the oxidation process of biogenic HCs by O(3P) (Paulson et al., 1992;Alvarado et al., 1998) was included to synchronize with the oxidation path in SAPRC."

9.  Line 148: It could be helpful to include the C* saturation vapor pressure equivalents for the vapor pressures listed here.
    **Response:** The saturated concentration (C*) of each lumping species can be estimated from vapor pressure of each lumping species by the following equation
    $$C^* = \frac{MW_{or}\gamma_{or,i}p_{L,i}^\circ}{RT}$$
    In the absence of inorganic seed aerosol, gas-partitioning of gas products is processed in organic phase. In general, the activity coefficient of organic species to SOA is nearly 1 (1998 Jang) and the MW$_{or}$ is not much changed at a given precursor system. However, $\gamma_{in,i}$ (activity coefficient of organic species or lumping species in salted aqueous seed) varies with lumping species and thus, C* is not interchangeable with vapor pressure. Equally saying that C* cannot be constant. In the UNIPAR model, C* is dynamically estimated by using the fixed vapor pressure and the estimated activity coefficient.

10. Line 149: Can you give quantitative descriptions for the reactivity scale?

    **Response:** We reported the mathematical description of model parameters including reactivity scales and physicochemical parameters, and mathematical equations for stoichiometric coefficients in the recently published paper by Yu et al (Supporting Information of ACP, 2022). In this manuscript, same reactivity scale has been used, and the physicochemical parameters and mathematical equations for stoichiometric coefficients are extended to biogenic HCs at four major oxidation paths to simulate day and night SOA mass as seen in Sect. S7.

11. Section 3.4 and Figure 1: I'm confused by which parts of the model mechanism uses MCM gas phase chemistry and which parts use SAPRC chemistry. Please clarify.
    **Response:** Please find the response to comment 7. The model parameters and the predetermined mathematical equations for stoichiometric coefficients for lumping bins were derived by using the product predicted from the semi-explicit mechanisms for the atmospheric oxidation of biogenic HCs. The model parameter and the equations are integrated to the predicted hydrocarbon consumption from any gas mechanisms. In order to support SOA formation in complex ambient air, model parameters and equations for stoichiometric coefficients were integrated with SAPRAC07TC. The hydrocarbon consumption predicted with both semi-explicit mechanisms and SAPRAC07TC well accords with that observed in chamber studies.

12. Fig. 2 (a): Why isn't there a line for $OM_P$ here?
    **Response:** There is as line for $OM_P$ in Fig. 2 (a) but it is too small to see. Isoprene SOA is mainly produced via aerosol phase reaction and thus, there is only a small amount of SOA produced via gas-particle partitioning.

13. Fig. 2: An indication of when t=0 is (e.g., injection time at night or sunrise in the daytime) would be helpful in these plots. Additionally, I recommend noting in the caption that meteorological parameters (e.g., temperature, relative humidity) varied between these cases in addition to the chemical parameters already noted. It might also be more appropriate to label the middle and right columns as "VOC + O3 + NO3" instead of just "VOC + O3".
    **Response:** In order to response to the reviewer, the x-axis and the label of the plots for the nocturnal SOA formation (Fig. 2) has been revised. For the daytime experiments in outdoor chamber, the use of local time is reasonable because gas oxidation and aqueous reactions of organic species are sensitive to local time closely associated with environmental conditions (i.e., temperature, humidity, and sunlight intensity). The environmental conditions are summarized in Table 1. The example of environmental conditions has been updated in the supporting information of the revised paper (Fig. S1).

14. Line 225: This conclusion about an increase in SOA formation because of aqueous reactions of ozonolysis products is unsupported. Can you include a figure that shows process-level detailed analysis of the model to support this?
    **Response:** The description of the role of aqueous reactions on SOA growth has been moved to section 4.2, which is related to sensitivity test of SOA model to various variables (Figure 4).
    L303: "In Fig.4, nocturnal SOA formation from the α-pinene ozonolysis increases by including acidic seed (wAHS). Thus, the small increase in chamber-generated SOA formation (Fig. 2(e) and (f)) by inorganic seed is mainly caused by the aqueous phase reaction of the ozonolysis products."

15. Line 242-243: Reporting the atmospheric range of pHs would be more useful if the manuscript also included information about the pH of the seeds used in the experiments.
    **Response:** The pHs of ammonium hydrogen sulfate (AHS) is influenced by humidity. Thus, pH of the wAHS in our simulation conditions has been added in the revised manuscript.
    L243: **"**To investigate the impact of aerosol acidity, the SOA formation is simulated in the presence of wAHS seed at pHs -1.5 and 0 corresponding to RHs 45% and 80%, respectively.**"**

16. Line 248-250: The sentence beginning with "Evidently…" is redundant.
    **Response:** The sentence beginning with "Evidently…" has been merged with the next sentence in the revised manuscript
    L267: "Evidently, our simulation suggested that the averaged molecular weight of β-caryophyllene oxidation products in highly reactive groups (VF or F) is 183.01 $g\ mol^{-1}$, while that from isoprene and α-pinene is 116.65 $g\ mol^{-1}$ and 143.29 $g\ mol^{-1}$, respectively."

17. Lines 258-259: The conclusion about the impact of photolysis of isoprene and α-pinene oxidation products on SOA production is unsupported.
    **Response:** The simulated properties (O:C ratio) of oxidation products from isoprene and α-pinene are added to the revised manuscript
    L261: "The UNIPAR model estimates the activity coefficient of lumping species in inorganic-salted aqueous phase by using lumping species' physicochemical parameters. Unlike isoprene (Beardsley and

Jang, 2016) or aromatic products (Han and Jang, 2022;Im et al., 2014;Zhou et al., 2019), α-pinene gas products are relatively hydrophobic and thus, their solubility is low in the aqueous phase with their large activity coefficients. Evidently, α-pinene SOA yields the lower O:C ratio (~0.56) than isoprene SOA (~ 1.1) in the model."

18. Lines 261-263: Why is the high reactivity relevant here?
**Response:** In general, photolysis products are multifunctional and highly reactive for heterogeneous reaction. In our model, oligomerization of products are governed by the $2^{nd}$ order reaction. The reaction rate of aerosol phase reactions increases with increasing concentrations in aerosol phase. In general, the product with a higher molecular weight has a low volatility and increase the concentration in aerosol phase due to a greater partitioning constant. According to the molecular weight of the β-caryophyllene oxidation products, the reaction rate of products in highly reactive group can efficiently yield SOA. Unlike β-caryophyllene, isoprene or α-pinene photolysis products are more volatile with a low molecular weight and thus, the low product concentration in aerosol phase can be inefficient to form SOA.

19. Lines 283-285: The speculation presented here can be supported by the modeling work in this paper. Including supporting plots/calculations would be useful.
**Response:** In order to response to the reviewer, Figure S1 has been added to the revised manuscript.
L294: "Figure S1 illustrates the stoichiometric coefficients of the two lumping species (low volatility species in group 2S and the medium reactivity species (one aldehyde group) in group 4M), which increase with increasing $NO_x$ and can significantly contribute to SOA mass. These two species originate from the reaction of a nitrate radical with the ozonolysis products."

20. Line 316: Is the difference between nighttime and daytime SOA for the same amount of initial VOC? Does this statement account for variation in emissions / PBLH between day and night?
**Response:** The sentence has been revised to provide clear point.
L326: "Figure 5 shows that isoprene and α-pinene produce more SOA mass by nighttime chemistry than daytime chemistry."
The objective of Figure 5 is to show the sensitivity of biogenic SOA to temperature. SOA yields were simulated under the same amount of HC consumption in the box model.

21. Line 320: Note that $NO_3$ can also be lost at night by reaction with fresh NO emissions.
**Response:** As referee mentioned, $NO_3$ can react with NO and produce $NO_2$, which can react with $O_3$ to reproduce $NO_3$ radical again. Thus, this information has been added to the revised manuscript.
L329: "At night, $NO_3$ radicals can react with NO to form $NO_2$, but $NO_3$ radicals can be regenerated via the reaction of $NO_2$ and $O_3$. In daytime, the role of NO3 on SOA formation can be minimal owing to its rapid photolysis (e.g., lifetime = 5s) (Magnotta and Johnston, 1980)."

22. Lines 371-373: How does the ratio of alpha-pinene to gasoline correspond to that in the atmosphere?

**Response:** For our chamber studies, gasoline total carbon concentrations were nearly 3000 ppbC and those of α-pinene were about 800 ppbC (80 ppb). In the urban aera, the emissions are dominated by anthropogenic sources.

23. Lines 383-385: Can you provide a plot to support the conclusion of this sentence, or at least a quantification of the conclusion presented here?

**Response:** To response to the referee, Figure S7 has been added to the revised manuscript.

24. Fig 3(f): Why is there no dotted black line in Figure 3(f)?
**Response:** A black dotted line is overlapped with a red dotted line in Fig. 3(f). This information has been added to the revised manuscript.
L233: "In Fig. 3(f), the difference in $OM_P$ between NS and SA conditions was not appeared."

25. Fig 4: Because RH=45% is right around the efflorescence point for ammonium sulfate, do these high and low RH cases actually probe significant differences in aerosol liquid water content? Also, for clarity, I recommend noting in the caption that this figure presents modeling results.
**Response:** In order to simulate the impact of RH on SOA formation in the presence of wet-inorganic aerosol, RHs were higher than ERH. RHs at 45% and 80% were chosen to have a large gap. In the UNIPAR model, the aerosol water content is estimated for a wide variety range of RH under varying seed compositions. Below ERH, the aerosol water content turns to be zero allowing no aqueous reactions.
A sentence in the figure caption has been modified to clarify that the Fig.4 is the simulation result.
"The simulated potential SOA yield from each oxidation path from the given HC consumption under (a) high $NO_x$ level ($HC/NO_x$ = 3 ppbC/ppb) and (b) low $NO_x$ level ($HC/NO_x$ = 10 ppbC/ppb)."

26. Fig 7: What is driving the oscillations in the fraction of SOA from alpha pinene (solid red line)?
**Response:** Please attention to the y-scale for the α-pinene SOA mass fraction of total SOA. The window of the y-scale was narrow down between 0.9 and 1.0. The actual size of oscillations is nearly negligible. The tiny change in $OM_P$ (SOA mass from partitioning) is caused by the small change in temperature and humidity inside the UF-APHOR chamber during the nighttime experiment.

**Comments to the author**:
Dear authors,

After revision, the manuscript has improved. The second round of referee comments, however, reveals that the following criteria for publication in ACP are not met:

- The scientific methods are not clearly outlined.
- Not all conclusions are adequately supported by material presented in the manuscript.

I would like to highlight two comments by Anonymous Referee #3, which I find most critical regarding these two criteria.

**[1] Anonymous Referee #3: "[...] many of the mechanistic conclusions stated in the paper are backed up only by previous literature, rather than a detailed analysis of the authors' modeling work" and "including some process-level modeling results to support the stated conclusions would greatly improve the science presented".**

> Editor's comment: I agree with the referee with this assessment. In the following, I am listing further references to text in the manuscript, which I think could benefit from showing how the conclusion was derived from the material presented in this manuscript.

**1-1. L.224:** "The small increase in SOA formation by inorganic seed is mainly caused by the aqueous phase reaction of the ozonolysis products" – Please indicate how this conclusion was derived from the material presented in this manuscript (e.g. in a sensitivity analysis regarding the aqueous phase chemistry).

**Response:** Please find the response to comment 14 from reviewer 3.  This sentence has been moved to the sensitivity section to indicate that the increase of nocturnal SOA formation by acidic seed is due to the aqueous phase reaction of ozonolysis products.

L303: "In Fig.4, nocturnal SOA formation from the α-pinene ozonolysis increases by including acidic seed (wAHS). Thus, the small increase in chamber-generated SOA formation (Fig. 2(e) and (f)) by inorganic seed is mainly caused by the aqueous phase reaction of the ozonolysis products."

**1-2. L.251:** "Low volatile products form from the autoxidation of ozonolysis products as reported in the previous studies." – Please indicate how this conclusion was derived from the material presented in this manuscript (e.g. by showing and comparing the concentration of autoxidation products over time).

**Response**: Please find the sentence below,

L225: "The importance of autoxidation products on terpene SOA formation has been demonstrated in the previous study by Yu et al for daytime chemistry (Yu et al., 2021). At night, the contribution of autoxidation on ozonolysis SOA depends on the concentration of α-pinene. For example, the attribution of autoxidation on SOA mass in experiment AP01 (Table 1) was nearly 15%, but it can increase due to gas-particle partitioning of non-autoxidation products onto SOA mass originating autoxidation products."

**1-3. L. 257:** "… α-pinene gas products are relatively hydrophobic and thus, less soluble in the aqueous phase" – Please indicate how this conclusion was derived from the material presented in this manuscript (e.g. by providing a list of physicochemical parameters).

**Response:** Please find the response for the comment 17 from the referee 3. The simulated properties (O:C ratio) of oxidation products from isoprene and α-pinene are added to the revised manuscript

L261: "The UNIPAR model estimates the activity coefficient of lumping species in inorganic-salted aqueous phase by using lumping species' physicochemical parameters. Unlike isoprene (Beardsley and Jang, 2016) or aromatic products (Han and Jang, 2022;Im et al., 2014;Zhou et al., 2019), α-pinene gas products are relatively hydrophobic and thus, their solubility is low in the aqueous phase with their large activity coefficients. Evidently, α-pinene SOA yields the lower O:C ratio (~0.56) than isoprene SOA (~1.1) in the model."

**1-4. L. 259:** "In case of β-caryophyllene, even after the photodegradation the product volatility is still low enough to significantly partition to the aerosol phase and heterogeneously form SOA." – Please indicate how this conclusion was derived from the material presented in this manuscript (e.g. by providing physicochemical parameters of β-caryophyllene photodegradation products).
**Response:** As an example of the physicochemical parameters of β-caryophyllene products, averaged MW of reactive groups were given in the following sentences.

L268: "In case of β-caryophyllene, even after the photodegradation the product volatility is still low enough to significantly partition to the aerosol phase and heterogeneously form SOA. Evidently, the simulated molecular weight of β-caryophyllene oxidation products with high reactivity is generally higher than that in isoprene or α-pinene products. For example, the averaged molecular weight of β-caryophyllene oxidation products in highly reactive groups (VF or F) is 183.01 $g\ mol^{-1}$, while that from isoprene and α-pinene is 116.65 $g\ mol^{-1}$ and 143.29 $g\ mol^{-1}$, respectively."

**1-5. L296:** "The large molecules originating from β-caryophyllene oxidation might have a poor solubility in aqueous phase, weakening the impact of aerosol acidity on OMAR." – As this is a model result, speculation is not needed. Please indicate how this conclusion was derived from the material presented in this manuscript (e.g. by providing physicochemical parameters of β-caryophyllene oxidation products).

**Response:** As an example of the physicochemical parameters of β-caryophyllene products, averaged MW of reactive groups were given in the following sentences.

L307: "The large molecules originating from β-caryophyllene oxidation might have a poor solubility in aqueous phase, weakening the impact of aerosol acidity on OM$_{AR}$. For example, the simulated O:C of β-caryophyllene SOA is estimated as ~0.27 under the chamber conditions in Fig. 2 (i)."

**[2] Anonymous Referee #3: "Section 3.4 and Figure 1: I'm confused by which parts of the model mechanism uses MCM gas phase chemistry and which parts use SAPRC chemistry. Please clarify."**

>Editor's comment: I agree with the referee with this assessment. To remedy this comment, I would suggest that the authors provide a more complete description of the model. In addition to the specific comments of Anonymous Referee #3, I added specific comments below, which I hope will prove helpful.

**2-1. L. 78:** "Lumping species and their stoichiometric coefficient and physicochemical parameters from the explicit gas mechanisms were individually obtained from the four different oxidation pathways with OH radicals, O$_3$, NO$_3$ radicals, and O($^3$P)." – How was lumping achieved?

The authors state that gas-oxidation chemistry was "explicitly processed" (l. 135) and MCM gas-phase chemistry was used by "individually processing" (l. 143). This explanation seems insufficient to evaluate the scientific method. Please indicate the detailed actions and methods used. See also the most recent comments by Anonymous Referee #1 regarding explicitness of the mechanism.

**Response:** The more information about the lumping species are added in the Sect. S7. Reactivity scales of the oxidation products are determined according to the number of reactive functional groups in their chemical structure. The reactivity bins used in UNIPAR are very fast (VF, α-hydroxybicarbonyls and tricarbonyls), fast (F, 2 epoxides or aldehydes,), medium (M, 1 epoxide or aldehyde), slow (S, ketones), partitioning only (P) and multi-alcohol (MA, 3 or more alcohols) (Beardsley and Jang, 2016;Yu et al., 2021;Zhou et al., 2019). The explicit chemical structures of oxidation products from isoprene, α- pinene, and β-caryophyllene are summarized in Table S5-S7 as examples.

**2-2.** Can the authors list all lumping species, stoichiometric coefficients and physicochemical parameters in the supplement? Can the authors add more detail how these were obtained (e.g. fitting by hand, fitting algorithm)?
**Response:** The physicochemical parameters of lumping specie and the equations to estimate stoichiometric coefficients has been included in Sect S7. Additionally, examples of the products in lumping bins have also been represented in Table S5-S7 of the revised manuscript.

**2-3.** The additional mechanisms (Section S1.2, Schemes S1-S4) seem not included in Table S1. What are the rate coefficients of the additional reactions in schemes S1-S4? I would suggest to provide a full list of all chemical reactions of the model in Table S4. The gasoline oxidation mechanism is also absent from Tab. S1.

**Response:** Please find the response to comments 7 and 11 from reviewer 3.

"The model parameters and the predetermined mathematical equations for stoichiometric coefficients for lumping bins were derived by using the products predicted from the semi-explicit mechanisms for the atmospheric oxidation of biogenic HCs. In this study, semi-explicit gas mechanisms were improved to predict low volatility products by the addition of autoxidation mechanisms and the combination of nitrate-originating peroxy radicals to the preexisting explicit gas mechanisms. Those additional mechanisms were illustrated in Schemes S1-S3 of Section S1.1. The model parameter and the equations are integrated to the predicted hydrocarbon consumption from any gas mechanisms. In order to support SOA formation in complex ambient air, model parameters and equations for stoichiometric coefficients were integrated with SAPRAC07TC (Table S4 in Section S1.2)." (see Sect. S1)

In order to respond to the comment, the mechanisms associated with Schemes S1-S3 were summarized in Table S1-S3. The format of mechanisms follows in those in the MCM. The oxidation mechanisms of aromatic HC which is related to the gasoline composition are added to the Table S4.

---

## Author Response (AR3)

**Editor's comment**

**Thank you very much for addressing most of the reviewer's comments, the manuscript is greatly improved. After minor revisions regarding previous referee comments, I am happy to accept the manuscript for publication.**

We would like to thank the editor for your time and thoughtful comments on this manuscript. Your comments are repeated below followed by our response to each comment.

**Referee #3** - Similarly, the abstract is very long-winded and makes it hard to understand the science question that the authors are trying to answer. I recommend revising the abstract to focus more on the big picture impact of this work rather than a list of detailed conclusions.
**Response:** The abstract has been revised based on the referee's comment.
**Review by Editor**: The track-changes file shows hardly any changes to the abstract, please address the reviewer's comment.
**Response to Editor**: To make the abstract clear, some sentences were modified, removed, or added in the revised manuscript. Changes are marked as red in the manuscript.

**Referee #3** - Line 149: Can you give quantitative descriptions for the reactivity scale?
**Response**: We reported the mathematical description of model parameters including reactivity scales and physicochemical parameters, and mathematical equations for stoichiometric coefficients in the recently published paper by Yu et al (Supporting Information of ACP, 2022). In this manuscript, same reactivity scale has been used, and the physicochemical parameters and mathematical equations for stoichiometric coefficients are extended to biogenic HCs at four major oxidation paths to simulate day and night SOA mass as seen in Sect. S7.
**Review by Editor**: Please provide a reference to Yu et al. around line 149.
**Response to Editor**: The reference has been added in the revised manuscript.

**Referee #3** - Section 3.4 and Figure 1: I'm confused by which parts of the model mechanism uses MCM gas phase chemistry and which parts use SAPRC chemistry. Please clarify.
**Response**: Please find the response to comment 7. The model parameters and the predetermined mathematical equations for stoichiometric coefficients for lumping bins were derived by using the product predicted from the semi-explicit mechanisms for the atmospheric oxidation of biogenic HCs. The model parameter and the equations are integrated to the predicted hydrocarbon consumption from any gas mechanisms. In order to support SOA formation in complex ambient air, model parameters and equations for stoichiometric coefficients were integrated with SAPRC07TC. The hydrocarbon consumption predicted with both semi-explicit mechanisms and SAPRC07TC well accords with that observed in chamber studies.
**Review by Editor**: I would like to reiterate on referee #3's question. When was MCM used and when was SAPRC used? Please refer to any previously published work here when determination of mathematical equations for stoichiometric coefficients for lumping bins was performed for a previous study and please outline the exact methodology (incl. showing calculation results) when it was newly performed for this study.

**Response to Editor:** The MCM mechanisms and additional explicit mechanisms are simulated to produce explicit products. The resulting products are classified into lumping species bins based on volatility-reactivity. These gas mechanisms are also simulated to build the model equations for lumping species' stoichiometric coefficients as a function of HC ppbC/NOx ppb ratios (ranging from 1 to 50) and gas products' aging, and yield lumping species' physicochemical parameters that are used to process multiphase partitioning and in-particle phase reaction. The resulting model parameters and pre-determined equations

can be applied to any gas mechanisms, which can predict the consumption of hydrocarbons and the production of $RO_2$ and $HO_2$ at given NOx levels. Thus, the UNIPAR model can be suitable to be simulated even in regional models that generally use Carbon Bond mechanisms or SAPRAC. In our study, the UNIPAR model was coupled with SAPRC07TC to perform the prediction of SOA formation.